# Automata-Conditioned Cooperative Multi-Agent Reinforcement Learning

Beyazit Yalcinkaya [1]   Marcell Vazquez-Chanlatte [2]   Ameesh Shah [1]   Hanna Krasowski [1]   Sanjit A. Seshia [1]

## Abstract

We study learning multi-task, multi-agent policies for cooperative, temporal objectives, under centralized training, decentralized execution. In this setting, using *automata* to represent tasks assigned to agents enables breaking down a team-level objective into simpler, smaller sub-tasks. However, existing approaches remain sample-inefficient and are limited to the single-task case, requiring retraining policies for each new task. In this work, we present *Automata-Conditioned Cooperative Multi-Agent Reinforcement Learning* (ACC-MARL), a framework for learning task-conditioned, decentralized team policies. We identify challenges to the feasibility of ACC-MARL, propose solutions, and prove that our approach is optimal. We further show that learned value functions can be used to assign tasks optimally at test time. Experiments demonstrate emergent task-aware, multi-step coordination among agents, such as pressing a button to unlock a door, holding the door, and short-circuiting tasks. Our code is available at https://github.com/rad-dfa/acc-marl.

## 1. Introduction

A major challenge in *Multi-Agent Reinforcement Learning* (MARL) is generalizing to a priori unknown tasks assigned at runtime, without retraining. In this multi-task setting, achieving *cooperative* team-level objectives becomes even more difficult when agents execute decentralized policies without explicit communication. In such cases, each agent must reason about the tasks assigned to all agents to make optimal decisions toward completing both its own task and the team's objectives. In turn, the representation of tasks becomes crucial for generalization and optimal team behavior.

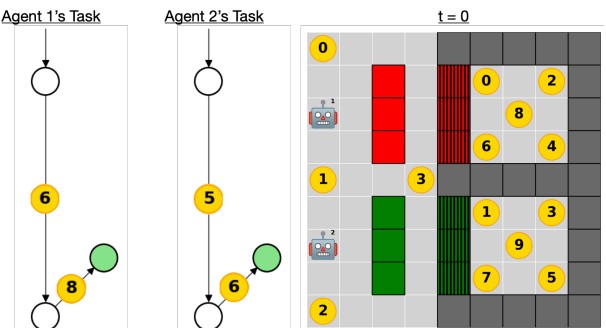

*Figure 1.* Motivating example **Buttons-2**—details are below.

*Formal specifications* have been proposed as a means for defining tasks for policies due to their well-defined operational semantics and concise encoding of long-horizon, temporally extended objectives (Icarte et al., 2022; Vaezipoor et al., 2021; Hasanbeig et al., 2020; Jothimurugan et al., 2019; Shah et al., 2025b; Yalcinkaya et al., 2024). In *cooperative MARL*, their *compositional* nature allows us to break down complex tasks into simpler, smaller ones while satisfying the overall task (Neary et al., 2021; Smith et al., 2023; Shah et al., 2025a). However, existing methods are sample inefficient and confined to a single fixed task.

In this work, we present *Automata-Conditioned Cooperative MARL* (ACC-MARL), a framework for learning multi-task, multi-agent policies where objectives are represented as *Deterministic Finite Automata* (DFAs). Our approach leverages recent work on *automata embeddings* for goal-conditioned RL (Yalcinkaya et al., 2024; 2025) to provide meaningful task representations for conditioning decentralized MARL policies, enabling efficient transfer of knowledge across semantically similar tasks. We motivate ACC-MARL and the challenges involved in its feasibility in the following.

**Motivating Example.** Consider the game in Figure 1 with two agents and many tokens scattered across two rooms. To open the doors of a room—shown with striped colored cells, an agent needs to stand on a corresponding button—shown with the same colored cells. At the beginning of the game, tasks that involve visiting these tokens are assigned to the agents. For example, "reach token 6 and then 8" and "reach token 5 and then 6" could be assigned to agents 1 and 2, respectively. *The goal is to complete all tasks successfully.* Due to the environment dynamics, agents must cooperate

[1]Department of Electrical Engineering and Computer Sciences, University of California, Berkeley, USA [2]Independent Researcher, CA, USA. Correspondence to: Beyazit Yalcinkaya <beyazit@berkeley.edu>.

*Proceedings of the 43rd International Conference on Machine Learning*, Seoul, South Korea. PMLR 306, 2026. Copyright 2026 by the author(s).

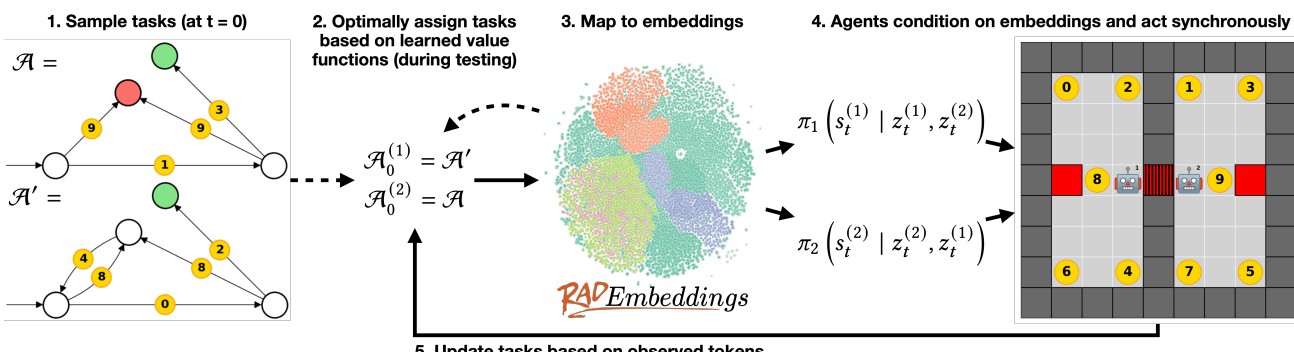

*Figure 2.* An overview of ACC-MARL with **Rooms-2** environment given on the right. During training, we sample tasks from a prior and randomly assign them. At each step, tasks are mapped to RAD Embeddings and passed to decentralized policies. Each agent conditions on these embeddings and synchronously predicts its action. At test time, we use learned value functions to assign tasks optimally.

and make coordinated moves to enter and exit the rooms. Moreover, each agent needs to know the other's task along with its own to decide *whether* and *how* to help one another.

We identify three main challenges to ACC-MARL in this setting. First, conditioning on temporal tasks requires learning history-dependent policies so that agents can track their progress, e.g., in Figure 1, agents need to infer previously-reached tokens. This can lead to sample inefficiency and result in suboptimal policies, as our ablation study in Section 5.1 shows. Second, the game objective provides a sparse reward signal, i.e., "did the team complete all tasks, or not?" For instance, agents in Figure 1 need to reach four tokens in the correct order before getting non-zero rewards. This results in a *credit assignment problem* (Agogino & Tumer, 2004), which makes it difficult for agents to discern how their individual behaviors contribute to the overall reward. Finally, generalizing to a large class of temporal tasks requires learning semantically meaningful latent task representations, e.g., we want agents in Figure 1 to handle various tasks assigned at runtime and transfer their skills across different task classes. However, learning latent representations for assigned DFAs while simultaneously learning policies conditioned on these representations can be a performance bottleneck, as we empirically show in Section 5.1.

This work introduces ACC-MARL and presents a novel approach addressing each of these challenges. For history dependency, we utilize operational semantics of DFAs and, at each step, augment agents' observations with the latest DFA representations of given tasks to provide each agent with information about how much progress has been made toward the assigned tasks. We employ potential-based reward shaping (Ng et al., 1999; Devlin & Kudenko, 2011) to address the credit assignment issue, rewarding each agent for completing its own sub-task while still learning optimal team behaviors. To overcome the representation bottleneck, we use RAD Embeddings (Yalcinkaya et al., 2024; 2025), pretrained DFA embeddings that distinguish distinct tasks

and encode semantic similarities across a large class of temporal tasks. Furthermore, we show that, after training, the learned value functions can be utilized to find optimal task assignments to agents at test time. Finally, we present an empirical evaluation demonstrating the efficacy of our approach. See Figure 2 for a high-level overview.

**Contributions.** We summarize our contributions as follows: (i) we introduce Automata-Conditioned Cooperative MARL (ACC-MARL), a framework for learning multi-task, multi-agent, ego-centric policies; we address challenges to its feasibility and prove that our approach is optimal; (ii) we show that the value functions of learned ACC-MARL policies can be used for assigning tasks optimally; (iii) we implement a toolchain in JAX (Bradbury et al., 2018), which enables parallelized operations on DFAs and can be used as a Python package to work with DFA tasks and to learn DFA-conditioned policies in other applications; (iv) we present an empirical evaluation of our framework, demonstrating its efficacy and qualitatively analyzing learned agent behaviors.

## 2. Preliminaries

We assume the (unknown) environment is a *Markov game*.

**Definition 2.1** (Markov Game)**.** A **Markov game** with $n$ agents is a tuple $\mathcal{M} = \langle S, A, P, \iota \rangle$, where $S = S_1 \times \cdots \times S_n$ is the joint set of states, $A = A_1 \times \cdots \times A_n$ is the joint set of actions, $P : S \times A \to \Delta(S)$ is the transition probability function, and $\iota \in \Delta(S)$ is the initial state distribution. We assume that each $S_i$ encodes the global state of the game, but from agent $i$'s point of view. $\tau \in S^*$ is called a **trace**, and we use $\tau_i \in S_i^*$ to indicate a trace of agent $i$. Episodes are assumed to be finite, and $T$ denotes the finite horizon.

Here, a reward specific to the Markov game is not specified. Instead, tasks are assigned at runtime in the form of *Deterministic Finite Automata* (DFAs), and rewards are given based on whether all tasks are completed, e.g., Figures 1 and 2. Therefore, we continue with our task model next.

**Definition 2.2** (Deterministic Finite Automaton). A **Deterministic Finite Automaton** (DFA) is a tuple $\mathcal{A} = \langle Q, \Sigma, \delta, q_0, F \rangle$, where $Q$ is the finite set of states, $\Sigma$ is the finite alphabet, $\delta : Q \times \Sigma \to Q$ is the transition function, where $\delta(q, \sigma) = q'$ denotes a transition to $q' \in Q$ from $q \in Q$ on $\sigma \in \Sigma$, $q_0 \in Q$ is the initial state, and $F \subseteq Q$ is the set of final states. The semantics of a DFA is defined by its set of final states $F$ and its extended (lifted) transition function $\delta^* : Q \times \Sigma^* \to Q$, where

$$\delta^*(q, \varepsilon) \triangleq q \ \text{ and } \ \delta^*(q, \sigma w) \triangleq \delta^*(\delta(q, \sigma), w).$$

If $\delta^*(q_0, w) \in F$, then $\mathcal{A}$ **accepts** $w$, i.e., $w \models \mathcal{A}$. If $\delta^*(q_0, w) \notin F$, then $\mathcal{A}$ **rejects** $w$, i.e., $w \not\models \mathcal{A}$. We assume that once a DFA accepts a prefix, a suffix cannot change the acceptance decision, i.e., $\forall q \in F, \forall \sigma \in \Sigma, \delta(q, \sigma) = q$.

DFAs can be reduced to a canonical form (up to an isomorphism) through minimization (Hopcroft, 1971), denoted by $\min(\mathcal{A})$. We write $\mathcal{A}_\top$ for the single-state accepting DFA. The *progression* of $\mathcal{A}$ by a word $w \in \Sigma^*$ is defined as:

$$\mathcal{A}/w \triangleq \min\left(\langle Q, \Sigma, \delta, \delta^*(q_0, w), F \rangle\right),$$

i.e., read the word and minimize the DFA. Given a set of DFAs $D$, we write $\mathbf{A} \in D^n$ to denote an $n$-tuple of DFAs.

We relate traces of the environment to DFAs through a *labeling function*, mapping states to alphabet symbols. Given a trace $\tau = s_0, \dots, s_k \in S^*$ and a labeling function $L : S \to \Sigma$, we write $\tau \models_L \mathcal{A}$ to denote $L(s_0), \dots, L(s_k) \models \mathcal{A}$.

# 3. ACC-MARL

This section presents our theoretical framework—see Figure 2 for an overview. First, we state the ACC-MARL problem and discuss the main challenges to its feasibility.

## 3.1. Problem Statement

We start with formally stating the ACC-MARL problem.

**Problem 3.1** (Automata-Conditioned Cooperative Multi-Agent Reinforcement Learning (ACC-MARL)). *Given a Markov game $\mathcal{M} = \langle S, A, P, \iota \rangle$ with $n$ agents, a finite set of DFAs $D$ over some shared alphabet $\Sigma$ with a prior distribution $\iota_D \in \Delta(D)$, and a labeling function $L_i : S_i \to \Sigma$ for $i \in [n]$, a decentralized policy for agent $i$ employs*

$$\pi_i : S_i^* \times D^n \to \Delta(A_i),$$

*where $S_i^*$ is the set of traces of agent $i$. The joint policy is*

$$\boldsymbol{\pi}(\tau, \mathbf{A}) = [\pi_1(\tau_1, \mathbf{A}), \dots, \pi_n(\tau_n, \mathbf{A})].$$

*The ACC-MARL problem is to find a joint policy $\boldsymbol{\pi}$ maximizing the probability of satisfying the conjunction of all DFAs*

*in $\mathbf{A} \sim \iota_D^n$ by navigating the underlying game $\mathcal{M}$, i.e.,*

$$J(\boldsymbol{\pi}) = \mathbb{P}_{\substack{\mathbf{A} \sim \iota_D^n \\ \tau \sim \mathcal{M}, \boldsymbol{\pi}, \mathbf{A}}} \left[ \bigwedge_{i=1}^n \tau_i \models_{L_i} \mathbf{A}[i] \right], \qquad (1)$$

*where $\tau$ is generated by conditioning $\boldsymbol{\pi}$ on $\mathbf{A}$ and running it in $\mathcal{M}$. The objective is to solve $\boldsymbol{\pi}^* \in \arg\max_{\boldsymbol{\pi}} J(\boldsymbol{\pi})$, i.e., maximize the probability of satisfying all DFAs in $\mathbf{A} \sim \iota_D$.*

Our goal is to solve Problem 3.1 in the centralized training, decentralized execution setting, where each agent observes the global state from its own point of view, along with all assigned DFAs, and locally predicts its next action. There are three main challenges to make this problem feasible.

1. **History Dependency.** Policies need to take the generated trace (history) to decide the current state, i.e., task progress, of each DFA in $\mathbf{A} \in D^n$. Thus, the augmented game given in Problem 3.1 is not a Markov game. In practice, this history dependency can result in suboptimal policies, as we empirically show in Section 5.1.

2. **Credit Assignment.** The objective given in Problem 3.1 defines a sparse reward solely based on the successful completion of the overall task, i.e., whether the team completed all assigned tasks or not. On the other hand, naively rewarding based on the individual tasks assigned to agents prevents cooperation. This makes it hard for agents to understand the impact of their own behaviors on the overall task objective, commonly known as the *credit assignment problem* (Agogino & Tumer, 2004).

3. **Representation Bottleneck.** Conditioning on DFAs couples control and representation learning, as policies need to learn latent DFA representations while simultaneously conditioning on them for control. This poses a challenge for scalability and generalization, as demonstrated in the single-agent setting (Yalcinkaya et al., 2024; 2025) and as Section 5.1 shows in the multi-agent case.

In the following, we present our approach for addressing these challenges and prove that it solves Problem 3.1.

## 3.2. Addressing History Dependency

We start with history dependency. First, recall that the formulation given in Problem 3.1 conditions policies on the DFAs assigned at the beginning of the episode and therefore requires agents to rely on the generated trace (history) to infer task progress. Second, observe that as we take transitions towards the accepting state of a DFA, the task changes, e.g., when agent 1 reaches token 6 in Figure 1, its DFA task becomes a smaller DFA, i.e., the DFA that says "reach token 8." We use this observation, along with the full-observability assumption for the underlying environment defined in Definition 2.1, to mitigate history dependency by updating agents' DFAs using given labeling functions and augmenting the state with the latest minimal DFAs.

Given a finite set of DFAs $D$ over some $\Sigma$ as in Problem 3.1, define its corresponding *DFA space* $\mathcal{D} \supseteq D$ as follows:

$$\mathcal{D} \triangleq \{\mathcal{A} \mid \exists \mathcal{A}' \in D, \exists w \in \Sigma^*, \mathcal{A} = \mathcal{A}'/w\}, \quad (2)$$

i.e., $\mathcal{D}$ contains all minimized sub-DFAs of $D$. Note that a similar notion has been introduced in the single-agent case (Yalcinkaya et al., 2025). Here, we extend it to the multi-agent setting. In Problem 3.1, tasks from $D$ are assigned to $n$ agents; therefore, the product $D^n$ has all such initial task assignments. Since $\mathcal{D}$ contains all minimized sub-DFAs of $D$, $\mathcal{D}^n$ contains all possible minimal sub-DFAs agents can see throughout an episode. Therefore, we can use this product space to expose the notion of task progress to agents.

A product DFA space $\mathcal{D}^n$ induces a deterministic MDP

$$\mathcal{M}_{\mathcal{D}^n} = \langle \mathcal{D}^n, \Sigma^n, T_{\mathcal{D}^n}, R_{\mathcal{D}^n}, \iota_D^n \rangle,$$

where

- $\mathcal{D}^n$, the product DFA space, is the set of states,
- $\Sigma^n$, the product alphabet, is the set of actions,
- $T_{\mathcal{D}^n} : \mathcal{D}^n \times \Sigma^n \to \mathcal{D}^n$ is the transition function s.t.

$$T_{\mathcal{D}^n}(\mathbf{A}, \boldsymbol{\sigma}) = \mathbf{A}/\boldsymbol{\sigma}, \quad (3)$$

  where $\mathbf{A}/\boldsymbol{\sigma} = [\mathbf{A}[i]/\boldsymbol{\sigma}[i]]_{i \in [n]}$ is element-wise progress,
- $R_{\mathcal{D}^n} : \mathcal{D}^n \times \Sigma^n \to \{0, 1\}$ is the reward function s.t.

$$R_{\mathcal{D}^n}(\mathbf{A}, \boldsymbol{\sigma}) = \begin{cases} 1 & \text{if } T_{\mathcal{D}^n}(\mathbf{A}, \boldsymbol{\sigma}) = \mathbf{A}_\top \\ 0 & \text{otherwise,} \end{cases} \quad (4)$$

  where $\mathbf{A}_\top = [\mathcal{A}_\top]_{i \in [n]}$ is a vector of $\mathcal{A}_\top$, and
- $\iota_D^n \in \Delta(D^n)$ is the prior distribution from Problem 3.1.

We take the cascade composition of the underlying Markov game $\mathcal{M}$ and $\mathcal{M}_{\mathcal{D}^n}$ and play the game in the product space $S \times \mathcal{D}^n$. In this new game, from state $(s_t, \mathbf{A}_t)$, given an action $a_t$, we (i) step $\mathcal{M}$ first, (ii) then label the resulting state using the element-wise labeling function $L(s_{t+1}) = [L_i(s_{t+1})]_{i \in [n]}$, (iii) step $\mathcal{M}_{\mathcal{D}^n}$ using these labels, (iv) give reward based on $R_{\mathcal{D}^n}$, and (v) finally, return the next state $(s_{t+1}, \mathbf{A}_{t+1})$. We denote this game with $\mathcal{M} \mid_L \mathcal{M}_{\mathcal{D}^n}$.

Note that this new formulation of the game introduces a negligible computational overhead compared to the original one. The DFA space formulation given in Equation (2) is never explicitly computed or stored in memory; instead, DFA spaces are implicitly defined by generating distributions—see Appendix B.2. At each step, we progress the DFAs as described in Equation (3), which involves updating the current DFA states using the observed symbols and then minimizing the resulting DFAs. The former step has $\mathcal{O}(1)$ complexity, i.e., read the next state from the transition table. For a DFA with $n$ states, the computational cost of DFA minimization is $\mathcal{O}(n \log n)$ when one uses Hopcroft's algorithm (Hopcroft, 1971), which is the best-known bound.

$\mathcal{M} \mid_L \mathcal{M}_{\mathcal{D}^n}$ is a Markov game as the policies have the latest minimal DFAs and consequently do not need the generated trace as input. Therefore, in this game, each agent $i$ employs

$$\pi_i' : S_i \times \mathcal{D}^n \to \Delta(A_i)$$

and the joint policy $\boldsymbol{\pi}'$ is obtained by synchronously running local policies. We use the following to solve $\mathcal{M} \mid_L \mathcal{M}_{\mathcal{D}^n}$:

$$J_\gamma'(\boldsymbol{\pi}') = \mathbb{E}_{\substack{s_0 \sim \iota \\ \mathbf{A}_0 \sim \iota_D^n}} \left[ \sum_{t=0}^{T} \gamma^t R_{\mathcal{D}^n}(\mathbf{A}_t, L(s_{t+1})) \right], \quad (5)$$

where $s_{t+1} \sim P(s_t, a_t)$, $a_t \sim \boldsymbol{\pi}'(s_t, \mathbf{A}_t)$, $\gamma \in [0, 1]$ is a discount factor, and $T$ denotes the finite horizon of the game. This reformulation explicitly tracks task progress by augmenting the state with the latest minimal DFAs, thereby making the game Markovian and mitigating history dependency. Next, we show that a policy maximizing Equation (5) is optimal with respect to Problem 3.1 as $\gamma \to 1^-$.

**Theorem 3.1.** *Maximizing $J_\gamma'(\boldsymbol{\pi}')$ solves Problem 3.1, i.e.,*

$$\lim_{\gamma \to 1^-} \max_{\boldsymbol{\pi}'} J_\gamma'(\boldsymbol{\pi}') = \max_{\boldsymbol{\pi}} J(\boldsymbol{\pi}),$$

*where $J(\boldsymbol{\pi})$ and $J_\gamma'(\boldsymbol{\pi}')$ are in Equations (1) and (5), resp.*

The proof is given in Appendix A. Theorem 3.1 states that our Markovian reformulation of the non-Markovian game given in Problem 3.1 has the same optimal policy. Therefore, we can use this reformulation to solve Problem 3.1, addressing history dependency. However, as we show in our ablation study in Section 5.1, using the Markovian formulation alone is not enough to learn optimal policies. The reward defined in Equation (4) is still sparse, returning non-zero rewards only when all agents complete their tasks. Therefore, in the following, we present our approach for shaping the reward while preserving the optimality of learned policies.

### 3.3. Addressing Credit Assignment

Our goal is to shape the sparse reward given in Equation (4), returning non-zero values only when all DFAs are satisfied, while still guaranteeing optimality with respect to Problem 3.1. To this end, we apply *potential-based reward shaping* (Ng et al., 1999; Devlin & Kudenko, 2011) by defining the potential function of each agent as the successful completion of its DFA. Formally, for an agent $i$, we define:

$$\Phi_i(\mathbf{A}) = \begin{cases} 1 \text{ if } \mathbf{A}[i] = \mathcal{A}_\top \\ 0 \text{ otherwise.} \end{cases}$$

We then use $\Phi_i$ to shape the reward of agent $i$ as follows:

$$R_{\text{PBRS}}^{(i)}(\mathbf{A}, \boldsymbol{\sigma}) = R_{\mathcal{D}^n}(\mathbf{A}, \boldsymbol{\sigma}) + F(\mathbf{A}, \boldsymbol{\sigma}) \quad (6)$$
$$F(\mathbf{A}, \boldsymbol{\sigma}) = \gamma \Phi_i(T_{\mathcal{D}^n}(\mathbf{A}, \boldsymbol{\sigma})) - \Phi_i(\mathbf{A}),$$

where $\gamma \in [0, 1]$ is the discount factor from Equation (5). Observe that agents still receive the same reward as Equation (4) when they complete all DFAs, but they also get a positive reward when they complete their own DFA tasks. We use the shaped reward $R_{\text{PBRS}}^{(i)}$ in the objective given by Equation (5) to train each policy $\pi_i'$. Maximizing this objective preserves optimality with respect to Problem 3.1, as the shaped reward is based on a state potential function, a result proved in (Devlin & Kudenko, 2011). By shaping the reward according to the successful completion of each agent's assigned DFA task, we provide denser feedback on how their behaviors contribute to the overall task, helping agents identify their roles and therefore addressing the credit assignment problem. As our ablation study in Section 5.1 shows, shaping the reward is crucial to learning optimal policies. However, in the same study, we also show that simultaneously learning latent DFA representations during training can be a performance bottleneck for teams with more than two agents. Therefore, we provide a solution to the representation bottleneck problem next.

### 3.4. Addressing Representation Bottleneck

Our ablation study in Section 5.1 shows that in four-agent games, given in Figures 6a and 6b in the Appendix, learning latent DFA representations during training can result in sub-optimal policies. In the single-agent case, decoupling representation and control learning has been shown to improve sample efficiency due to the large DFA classes considered (Yalcinkaya et al., 2024). Here, we extend this idea to MARL and represent DFAs using RAD Embeddings (Yalcinkaya et al., 2024; 2025), *provably correct pretrained DFA embeddings*. These latent representations enable skill transfer for downstream policies by encoding similarities across a large class of DFAs. They also uniquely represent distinct tasks, which is a crucial property used in the following.

Let $\Psi : \mathcal{D} \rightarrow \mathcal{Z}$ denote a pretrained encoder mapping DFAs in $\mathcal{D}$ to RAD Embeddings, i.e., latent DFA representations in $\mathcal{Z}$ as described in (Yalcinkaya et al., 2025)—see Appendix B.3 for more details on these embeddings. The encoder $\Psi$ guarantees that distinct DFAs are uniquely represented in the latent space. Formally, for all $\mathcal{A}, \mathcal{A}' \in \mathcal{D}$,

$$\min(\mathcal{A}) = \min(\mathcal{A}') \iff \Psi(\mathcal{A}) = \Psi(\mathcal{A}'), \quad (7)$$

i.e., two DFAs have the same embedding if and only if they are the same when minimized. As minimized DFAs are canonical task representations, if two DFAs are equal when minimized, then they represent the same task. Therefore, we can expose the product latent space $\mathcal{Z}^n$ to policies, instead of the product DFA space $\mathcal{D}^n$. Then, each agent $i$ employs

$$\pi_i'' : S_i \times \mathcal{Z}^n \rightarrow \Delta(A_i)$$

and the joint policy $\pi''$ is the synchronous composition of these local policies. We use the objective defined in Equation (5) with the shaped reward given in Equation (6) to learn $\pi''$. Note that $\Psi$ maps two DFAs to the same embedding if and only if they represent the same task, as stated in Equation (7). Therefore, the Markovian reformulation of Problem 3.1 given in Section 3.2, which is over the product DFA space $\mathcal{D}^n$, can be reformulated as one over the product latent DFA space $\mathcal{Z}^n$, with equivalent rewards and transition probabilities. Thus, the objective set for $\pi''$ is optimal with respect to Problem 3.1, which addresses the representation bottleneck problem while preserving optimality. As Section 5.1 demonstrates empirically, RAD Embeddings enable optimal policy learning in teams with more than two agents.

This concludes the details of our approach to make Problem 3.1 feasible. Next, we show that learned value functions can be used for assigning tasks optimally at test time.

### 3.5. Optimal Task Assignment

So far, we have assumed that the tasks in $\mathbf{A} \in D^n$ are assigned to agents, i.e., $\mathbf{A}[i]$ is the DFA of agent $i$. Now we show that solving Problem 3.1 provides a means for optimally assigning tasks if agents are allowed to share the outputs of their value functions at the episode start. Specifically, the learned value functions order task assignments with respect to the given objective and therefore can be used for finding optimal task assignments.

Let $V_i : S_i^* \times D^n \rightarrow \mathbb{R}$ denote the optimal value function of the decentralized policy $\pi_i : S_i^* \times D^n$ maximizing $J(\boldsymbol{\pi})$ as in Problem 3.1, and let $\mathbf{A} \in D^n$ be a DFA task assignment. Define $V : D^n \rightarrow \mathbb{R}$ as:

$$V(\mathbf{A}) \triangleq \mathbb{E}_{\tau \sim \mathcal{M}, \boldsymbol{\pi}, \mathbf{A}} \left[ \sum_{i=1}^{n} V_i(\tau_i, \mathbf{A}) \right].$$

We can use this as a proxy value function over task assignments and enumerate the Pareto frontier, i.e.,

$$\mathbf{A}^\star \in \underset{\mathbf{A}' \in \text{perm}(\mathbf{A})}{\arg\max} V(\mathbf{A}'),$$

where $\text{perm}(\mathbf{A})$ denotes all permutations of $\mathbf{A}$. $\mathbf{A}^\star$ is a Pareto optimal task assignment, as no agent's expected performance can be improved without degrading the overall performance, i.e., $V(\mathbf{A}^\star) \geq V(\mathbf{A}')$ for all $\mathbf{A}' \in \text{perm}(\mathbf{A})$. This ensures that we select from the set of non-dominated assignments, enabling optimal assignment of tasks to agents at test time. See Figures 2 and 6b for examples, where optimal task assignments allow agents to leverage their asymmetric conditions to improve team performance. Note that searching over all possible task assignments, i.e., $\text{perm}(\mathbf{A})$, could be intractable for large teams; however, teams of appropriate size could still benefit. Indeed, in Section 5.2, we show that our task assignment method improves team performance and yields higher empirical success probabilities. Observe that this result applies to the policies presented in Sections 3.2 to 3.4, as all are optimal with respect to Problem 3.1.

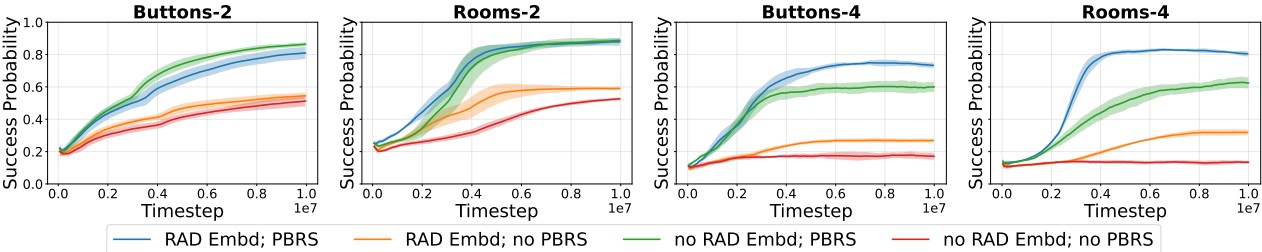

*Figure 3.* Success probabilities of learned policies throughout training, reported over 5 random seeds—shaded regions indicate standard deviation. "RAD Embd; PBRS" refers to Markovian policies conditioning on pretrained RAD Embeddings and trained with the shaped reward, i.e., the full solution proposed in Section 3. We present the results with the history-dependent baseline in Figure 8 in the Appendix.

## 4. Implementation

We implement a fully JAX-based toolchain[1] to enable efficient training. Experiments are conducted in a new environment called `TokenEnv`, a discrete, fully observable multi-agent environment, where agents observe the global state from their own perspective and must coordinate via buttons and doors; we consider both two- and four-agent variants with multiple layouts. All DFA-related operations are implemented in a native JAX package, `DFAx`, which also provides three task samplers: `Reach` DFAs (ordered token-reaching tasks), `ReachAvoid` DFAs (ordered reach tasks with unrecoverable avoid constraints), and `ReachAvoidDerived` (`RAD`) DFAs, which are obtained by randomly mutating `Reach` and `ReachAvoid` DFAs to induce richer branching task structure—see Appendix B.2 for details on these task classes. We deploy a pretrained and frozen Graph Attention Network (GATv2) (Brody et al., 2022) to map DFAs to RAD Embeddings, as described in (Yalcinkaya et al., 2025)—see Appendix B.3 for details on pretraining.

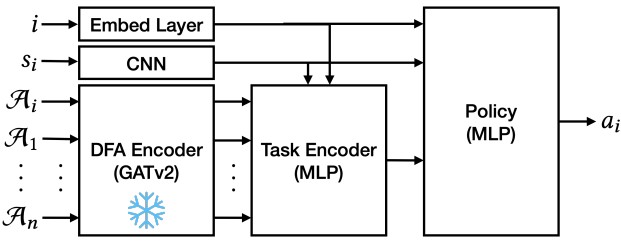

*Figure 4.* Policy architecture of an agent $i$, where Embed Layer is a learned lookup table, CNN is a convolutional neural network, MLP is a multilayer perceptron, and GATv2 is a graph attention network.

Each agent deploys the same decentralized policy presented in Figure 4, which takes the agent's ID, local observation, and the RAD Embeddings of all agents' current progressed DFAs as input and outputs an action. We use Independent Proximal Policy Optimization (de Witt et al., 2020; Lu et al., 2022; Rutherford et al., 2024) for training these policies. See Appendix B for more details on the implementation.

---

[1]All packages are available at https://github.com/rad-dfa.

## 5. Experiments

This section presents an empirical evaluation of the proposed framework. Our goal is to answer the questions listed below.

**(Q1)** *Does our approach make ACC-MARL feasible?*
**(Q2)** *Does it scale with an increasing number of agents?*
**(Q3)** *Do learned policies generalize?*
**(Q4)** *Do optimal task assignments improve performance?*
**(Q5)** *Do learned policies exhibit useful cooperative skills?*

We conduct an ablation study to answer **(Q1)**, learning policies for all variations discussed in Section 3 on `ReachAvoidDerived` (`RAD`) DFA tasks. To answer **(Q2)**, we train these policies in two- and four-agent layouts of `TokenEnv`. For **(Q3)**, we test the learned policies on `Reach`, `ReachAvoid`, and `RAD` DFAs. To further test the generalization capabilities of learned policies, we also test them on DFAs with more states than seen during training. We then compare the empirical success probabilities of agents under random and optimal task assignments to address **(Q4)**. Finally, we qualitatively analyze agent behaviors and identify learned skills to answer **(Q5)**.

### 5.1. Ablation Study

We first tackle **(Q1)** and **(Q2)**—short answers are below.

**(A1)** *The proposed approach makes ACC-MARL feasible.*
**(A2)** *ACC-MARL efficiently scales from two to four agents.*

Recall that in Section 3, three main challenges to Problem 3.1 are discussed: history dependency, credit assignment, and representation bottleneck; and three solutions are introduced: the Markovian reformulation, potential-based reward shaping (PBRS), and using pretrained RAD Embeddings, respectively. We train policies for all combinations of these solutions to identify their impacts. For the non-Markovian formulation of the game, we use an LSTM instead of the MLP Task Encoder given in Figure 4 so that policies can track task progress. For the second solution, we train policies with and without PBRS. Finally, we try both a pretrained (and frozen) DFA encoder and an untrained one.

| | | Success Probability | | | | | |
|---|---|---|---|---|---|---|---|
| Env | Policy | Reach | ReachAvoid | RAD | Reach (OOD) | ReachAvoid (OOD) | RAD (OOD) |
| Buttons-2 | RAD Embd; PBRS | $0.860 \pm 0.042$ 
 $0.869 \pm 0.041$ | $\mathbf{0.790 \pm 0.057}$ 
 $\mathbf{0.792 \pm 0.053}$ | $0.821 \pm 0.044$ 
 $0.825 \pm 0.045$ | $0.677 \pm 0.085$ 
 $0.670 \pm 0.084$ | $\mathbf{0.456 \pm 0.074}$ 
 $\mathbf{0.449 \pm 0.066}$ | $0.522 \pm 0.046$ 
 $0.520 \pm 0.044$ |
| | no RAD Embd; PBRS | $\mathbf{0.920 \pm 0.019}$ 
 $\mathbf{0.920 \pm 0.021}$ | $0.762 \pm 0.011$ 
 $0.764 \pm 0.013$ | $\mathbf{0.859 \pm 0.020}$ 
 $\mathbf{0.864 \pm 0.021}$ | $\mathbf{0.778 \pm 0.023}$ 
 $\mathbf{0.780 \pm 0.025}$ | $0.368 \pm 0.024$ 
 $0.375 \pm 0.017$ | $\mathbf{0.601 \pm 0.028}$ 
 $\mathbf{0.605 \pm 0.022}$ |
| Rooms-2 | RAD Embd; PBRS | $0.890 \pm 0.034$ 
 $0.919 \pm 0.032$ | $\mathbf{0.871 \pm 0.040}$ 
 $\mathbf{0.908 \pm 0.015}$ | $0.866 \pm 0.032$ 
 $0.895 \pm 0.023$ | $0.723 \pm 0.082$ 
 $0.759 \pm 0.085$ | $\mathbf{0.577 \pm 0.065}$ 
 $\mathbf{0.609 \pm 0.060}$ | $0.559 \pm 0.039$ 
 $0.574 \pm 0.037$ |
| | no RAD Embd; PBRS | $\mathbf{0.915 \pm 0.009}$ 
 $\mathbf{0.944 \pm 0.010}$ | $0.790 \pm 0.027$ 
 $0.838 \pm 0.028$ | $\mathbf{0.870 \pm 0.016}$ 
 $\mathbf{0.914 \pm 0.009}$ | $\mathbf{0.851 \pm 0.029}$ 
 $\mathbf{0.863 \pm 0.027}$ | $0.492 \pm 0.061$ 
 $0.523 \pm 0.064$ | $\mathbf{0.699 \pm 0.036}$ 
 $\mathbf{0.718 \pm 0.038}$ |
| Buttons-4 | RAD Embd; PBRS | $\mathbf{0.751 \pm 0.044}$ 
 $\mathbf{0.759 \pm 0.043}$ | $\mathbf{0.676 \pm 0.032}$ 
 $\mathbf{0.685 \pm 0.039}$ | $\mathbf{0.736 \pm 0.031}$ 
 $\mathbf{0.741 \pm 0.019}$ | $\mathbf{0.482 \pm 0.029}$ 
 $\mathbf{0.485 \pm 0.036}$ | $\mathbf{0.282 \pm 0.021}$ 
 $\mathbf{0.290 \pm 0.018}$ | $\mathbf{0.355 \pm 0.005}$ 
 $\mathbf{0.357 \pm 0.012}$ |
| | no RAD Embd; PBRS | $0.725 \pm 0.033$ 
 $0.727 \pm 0.044$ | $0.366 \pm 0.035$ 
 $0.366 \pm 0.034$ | $0.568 \pm 0.032$ 
 $0.578 \pm 0.026$ | $0.345 \pm 0.030$ 
 $0.347 \pm 0.022$ | $0.106 \pm 0.010$ 
 $0.109 \pm 0.009$ | $0.245 \pm 0.010$ 
 $0.247 \pm 0.011$ |
| Rooms-4 | RAD Embd; PBRS | $\mathbf{0.832 \pm 0.017}$ 
 $\mathbf{0.856 \pm 0.025}$ | $\mathbf{0.764 \pm 0.018}$ 
 $\mathbf{0.830 \pm 0.018}$ | $\mathbf{0.795 \pm 0.010}$ 
 $\mathbf{0.830 \pm 0.015}$ | $\mathbf{0.539 \pm 0.058}$ 
 $\mathbf{0.544 \pm 0.069}$ | $\mathbf{0.363 \pm 0.020}$ 
 $\mathbf{0.381 \pm 0.020}$ | $\mathbf{0.392 \pm 0.012}$ 
 $\mathbf{0.400 \pm 0.010}$ |
| | no RAD Embd; PBRS | $0.756 \pm 0.056$ 
 $0.813 \pm 0.047$ | $0.341 \pm 0.029$ 
 $0.427 \pm 0.030$ | $0.600 \pm 0.049$ 
 $0.667 \pm 0.033$ | $0.394 \pm 0.056$ 
 $0.413 \pm 0.051$ | $0.108 \pm 0.014$ 
 $0.121 \pm 0.014$ | $0.281 \pm 0.023$ 
 $0.302 \pm 0.012$ |

*Table 1.* Results are for random (top) and optimal assignments (bottom), and averaged over 5 seeds, each run for 1,000 episodes. Here, "RAD Embd; PBRS" refers to Markovian policies with PBRS and pretrained RAD Embeddings, i.e., the full solution in Section 3.

Figure 3 presents the success probabilities of learned policies throughout the training. Here, for instance, "RAD Embd; PBRS" refers to the case where we train a Markovian policy with pretrained RAD Embeddings and PBRS. We also present the same results in terms of discounted returns in Figures 9 and 10 in the Appendix. Note that in **Buttons-2** and **Rooms-2**, all history-dependent policies fail to achieve more than $0.5$ success probability. Therefore, to ease the exposition here, we report those results with history-dependent policies in Figure 8 in the Appendix.

Figure 3 shows that without PBRS, none of the policies can escape sub-optimal solutions, highlighting the impact of proper credit assignment. In **Buttons-2**, Markovian policies without pretrained RAD Embeddings, i.e., "no RAD Embd; PBRS," achieve a higher mean success probability and a lower variance than Markovian policies with pretrained RAD Embeddings, i.e., "RAD Embd; PBRS." In **Rooms-2**, RAD Embeddings provide lower variance even though policies without pretrained RAD Embeddings converge to the same success probability. On the other hand, in both four-agent environments, i.e., **Buttons-4** and **Rooms-4**, "RAD Embd; PBRS" policies achieve a higher success probability than "no RAD Embd; PBRS." More importantly, policies with pretrained RAD Embeddings demonstrate similar convergence behaviors in both two- and four-agent environments. This suggests that, rather than learning latent DFA representations during training, using pretrained RAD Embeddings is crucial to making ACC-MARL feasible, especially for larger teams and harder coordination problems. Overall, we conclude that our approach makes Problem 3.1 both feasible and scalable, answering **(Q1)** and **(Q2)**.

### 5.2. Evaluating Policies and Task Assignments

We continue with **(Q3)** and **(Q4)**—short answers are below.

**(A3)** *Learned policies exhibit generalization across tasks.*
**(A4)** *Optimal task assignments improve team performance.*

We take the best policies from Section 5.1, i.e., "RAD; PBRS" and "no RAD; PBRS," and test them for 1,000 episodes on various task classes. Notice that these policies are trained on RAD DFAs with at most 5 states. So, we test the policies on Reach, ReachAvoid, and RAD DFAs with at most 5 states, evaluating the performances of learned policies with respect to different task distributions. We also test these policies on out-of-distribution (OOD) DFAs, i.e., DFAs with at most 10 states, denoted by the "(OOD)" suffix.

The results are presented in Table 1, where the top line of each cell reports the results of random task assignments, and the bottom line is for the optimal ones. Overall, policies generalize to Reach and ReachAvoid DFAs. The policy without pretrained RAD Embeddings outperforms the one with RAD Embeddings on both Reach and RAD DFAs in **Buttons-2** and **Rooms-2**, whereas it falls short for ReachAvoid DFAs in these environments, suggesting that the notion of Avoid, i.e., the mission cannot be recovered once an avoid token is reached, is captured better by pretrained RAD Embeddings. On the other hand, in both four-agent environments, policies with pretrained RAD Embeddings outperform the baseline across the board. This suggests that the impact of pretrained RAD Embeddings manifests itself best with larger teams in harder environments. We see a similar pattern for OOD DFAs. Addi-

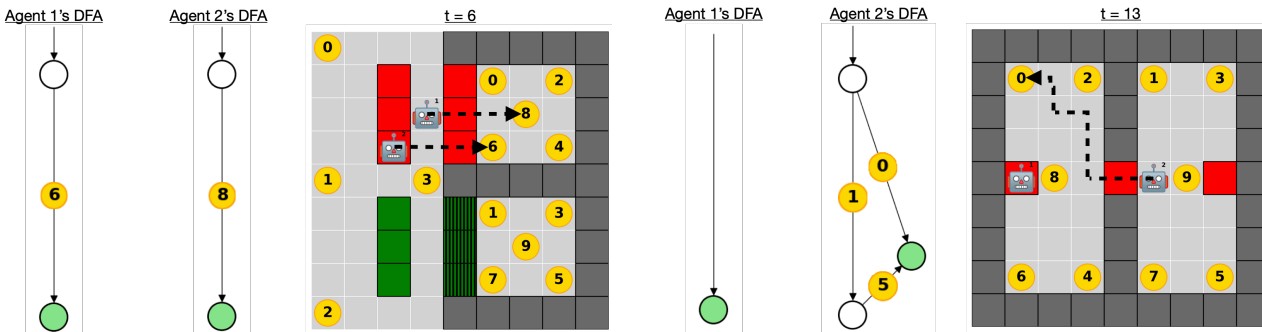

*(a)* Agents go to the same room by moving together and keeping the door open to save time, i.e., agent 1 *holds the door* for agent 2.

*(b)* Agents *short-circuit* the task, i.e., agent 1 opens the door and agent 2 completes its DFA by reaching one token instead of two.

*Figure 5.* A qualitative analysis of learned policies. See Figures 11 to 14 in the Appendix for longer traces.

tionally, for OOD DFAs, in two-agent environments, both policies demonstrate noticeable generalization on Reach tasks. Overall, we conclude that policies trained on RAD DFAs exhibit generalization, answering **(Q3)**.

Comparing top and bottom lines of each cell in Table 1, we see that optimally assigning tasks does not change the performance in **Buttons-2** and **Buttons-4**, as the agents are in almost symmetric states. On the other hand, in **Rooms-2** and **Rooms-4**, as expected, we see a performance improvement compared to random assignments. This confirms our observation in Section 3.5 that learned value functions can be used for computing optimal task assignments and therefore improve team performance, answering **(Q4)**.

### 5.3. Qualitative Analysis of Learned Policies

Finally, we answer **(Q5)**—the short answer is given below.

**(A5)** *Learned policies exhibit cooperative behaviors, such as pressing a button to unlock a door, holding the door, and helping other agents short-circuit their tasks.*

We conduct a qualitative analysis of the learned policies with pretrained RAD Embeddings and PBRS, i.e., "RAD Embd; PBRS" in Figure 3. We show that in **Buttons-2**, agents learn to *hold the door* for each other to save time, and in **Rooms-2**, agents learn to *short-circuit their DFAs* with the help of a *helper agent*. To ease the exposition, we present these behaviors in a compact form. A more detailed analysis is presented in Figures 11 to 14 in the Appendix.

Consider the case in Figure 5a for **Buttons-2**, where agent 1 is assigned a DFA that says "reach token 6," agent 2's DFA says "reach token 8," and the initial state of the environment is as in Figure 1, so the agents need to go to the same room. At $t = 6$, agents meet by the door, where agent 1 waits, and agent 2 presses a red button to open red doors—see state of the environment in Figure 5a. Then, instead of going into the room in turns, agents synchronously move towards the room, i.e., at $t = 7$, agent 1 is on the door, holding it,

and agent 2 is in front of the door. They keep moving in this manner until both agents are in the room and complete their tasks. This behavior shows that agents utilize the environment dynamics to accomplish their tasks optimally (see Figure 12 in the Appendix for another example).

For **Rooms-2**, consider the case given in Figure 5b, where agent 1 is assigned a trivially accepting DFA, i.e., agent 1 does not have a task—it is an helper agent, and agent 2's DFA says "reach token 0, or reach tokens 1 and 5 in that order," and the initial state of the environment is as given in Figure 2. At $t = 13$, given in Figure 5b, agent 1 is on the red button to unlock the door so that agent 2 can short-circuit its DFA by taking the shorter path, i.e., agent 2 can reach a single token in the other room instead of reaching two in its current room. From this point onward, agent 1 stays away from agent 2's path, and agent 2 reaches token 0 and completes its DFA. This behavior shows that if there are multiple ways to complete a DFA, agents learn to cooperate so that the task can be accomplished optimally (see Figure 14 in the Appendix for another trace of this behavior).

## 6. Related Work

**Multi-Task RL with Formal Specifications.** Formal specifications have become popular for task representation in multi-task RL due to their ability to define multiple tasks over a common alphabet of environmental events. Previous work has explored instructing a goal-conditioned policy to follow symbolically computed paths in automata (Jothimu-rugan et al., 2021; Qiu et al., 2023; Jackermeier & Abate, 2025; Guo et al., 2025). Others have proposed conditioning on temporal logic formulas (Vaezipoor et al., 2021) and automata (Yalcinkaya et al., 2023). Further results have shown that the graph structure of automata allows defining useful priors to facilitate generalization by pretraining automata embeddings and using them for downstream control (Yal-cinkaya et al., 2024; 2025). However, these efforts are limited to RL—we extend them to the multi-agent setting.

**MARL with Formal Specifications.** The use of formal specifications in MARL has been first studied under known environment dynamics (Karimadini et al., 2016; Schillinger et al., 2016; 2018). In model-free settings such as ours, different types of specifications have been used as a single fixed objective (Hammond et al., 2021; Sun et al., 2020; Terashima et al., 2024; Paul et al., 2024). Prior work has also explored hand-designing decompositions (Neary et al., 2021) and using heuristics (Smith et al., 2023) to assign tasks to agents in a team. Recent efforts have proposed simultaneously learning optimal task decompositions and policies to achieve a single fixed objective (Shah et al., 2025a). However, to the best of our knowledge, using formal specifications in the multi-task setting, i.e., a priori unknown tasks assigned at runtime, has not been considered.

**Multi-Task MARL.** Prior work in multi-task MARL takes a variety of approaches to learn policies that can generalize across tasks. Hierarchical methods have been studied for learning skill graphs (Zhu et al., 2025; Zhang et al., 2023), for separating the next goal allocation problem from goal-conditioned policies (Iqbal et al., 2022; Li et al., 2024), and for learning sub-task policies that are solutions of sub-MDPs (Wang et al., 2021; Matsuyama et al., 2025). Others have proposed factorized value functions (Iqbal et al., 2021) and transformers (Hu et al., 2021) to robustly generalize across tasks. Scheduling (Yu et al., 2023; Zhu et al., 2023) and knowledge distillation (Mai et al., 2024) have been shown to improve sample efficiency in multi-task MARL. This problem has also been considered under partial observability (Omidshafiei et al., 2017). To the best of our knowledge, we are the first to use formal specifications, i.e., structurally rich temporal task representations, in multi-task MARL.

## 7. Discussion

In this section, we discuss the benefits and limitations of the proposed framework, highlight important details, and propose potential directions for future work.

We start by highlighting the benefits of using DFAs as an instruction modality. First, their operational semantics enable efficient DFA progression as described in Section 3.2, which in turn allows learning Markovian policies. Second, the compositional nature of DFAs enables a complex task to be decomposed into simpler sub-tasks that can be assigned to individual agents, while still ensuring satisfaction of the overall specification (Neary et al., 2021; Smith et al., 2023; Shah et al., 2025a). Third, DFAs are expressive for finite-horizon behaviors: any finite behavior over an alphabet can be represented as a path in a DFA, and a set of such behaviors can be specified by encoding each finite behavior as a path and then minimizing the resulting DFA to get a compact task representation (Sipser, 1996). Thus, DFAs are crucial to the scalability and expressivity of our framework.

We continue with a discussion of the limitations and drawbacks of our framework. First, one major drawback of using DFAs is the assumption of a fixed labeling function per agent, which maps environment observations to alphabet symbols, as defined in Problem 3.1, and thus defines the semantics of DFA tasks. In our current setting, these labeling functions are known and remain fixed across training and evaluation. Extending our framework to support dynamic or learned labeling functions, i.e., an open or evolving alphabet where task semantics may change across episodes, is an important direction for future work, but it introduces additional challenges that warrant in-depth investigation in their own right and are therefore beyond the scope of this paper. Second, the proposed approach assumes full observability of the environment state; however, in certain application domains, it might not be possible to maintain this assumption. Therefore, another promising avenue for future work is to extend the proposed approach to partially observable settings. Third, as discussed in Section 3.5, the proposed approach for optimally assigning tasks at test time requires enumerating all assignments. Even though this is not a major issue for teams of appropriate size, it can be a bottleneck for large teams. To this end, RAD Embeddings and function approximation can be utilized to develop scalable learning-based solutions for the task assignment problem, which is another potential direction for future work. Lastly, our empirical evaluation in Section 5 is based solely on a discrete domain. Although this provides a controlled means for validating our framework, applications to continuous domains, high-fidelity simulators, and real robotic platforms remain open for further future investigations.

We end by emphasizing that our work is the first to use formal specification in multi-task cooperative MARL, with homogeneous agents. As such, there are no directly comparable baselines in the literature that address the same problem setting. In the context of multi-task MARL, a natural direction for future work is to consider the use of formal specifications in heterogeneous teams, where agents may have different action and observation capabilities, as well as in adversarial, mixed cooperative, and competitive settings.

## 8. Conclusion

This paper introduced ACC-MARL, a framework for task-conditioned multi-agent team policies. First, we addressed challenges to its feasibility and proved that our approach is optimal with respect to the stated problem. Second, we showed an optimal task assignment method using learned value functions. Third, we discussed our implementation, including a toolchain for incorporating automata tasks in other applications. Finally, we presented an empirical evaluation, highlighting the efficacy of ACC-MARL in learning multi-task, multi-agent, decentralized team policies.

## Acknowledgements

This work is partially supported by the DARPA ANSR and TIAMAT programs, by Nissan under the iCyPhy Center, and by NVIDIA's Academic Grant Program.

## Impact Statement

This paper presents work whose goal is to advance the fields of machine learning and reinforcement learning. There are many potential societal consequences of our work, none of which we feel must be specifically highlighted here.

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

# A. Proof of Theorem 3.1

**Theorem A.1.** *Maximizing $J'_\gamma(\pi')$ solves Problem 3.1, i.e.,*

$$\lim_{\gamma \to 1^-} \max_{\pi'} J'_\gamma(\pi') = \max_{\pi} J(\pi),$$

*where $J(\pi)$ and $J'_\gamma(\pi')$ are in Equations (1) and (5), resp.*

*Proof.* We first show that for every state of $\pi$, there exists a state of $\pi'$ with equivalent reward and transition probabilities. We then use this fact to prove the theorem.

Define a mapping $\mapsto: (S^* \times D^n) \to (S \times \mathcal{D}^n)$ such that

$$(s_0, \ldots, s_t, \mathbf{A}_0) \mapsto (s_t, \mathbf{A}_t), \tag{8}$$

where $\mathbf{A}_t = \mathbf{A}_0/L(s_0), \ldots, L(s_t)$. Write $J(\pi)$ in terms of its step reward by assuming all non-zero rewards are terminal:

$$
\begin{aligned}
J(\pi) &= \mathbb{P}_{\substack{\mathbf{A}_0 \sim \iota_D^n \\ \tau \sim \mathcal{M}, \pi, \mathbf{A}_0}} \left[ \bigwedge_{i=1}^n \tau \models_{L_i} \mathbf{A}_0[i] \right] \\
&= \mathbb{E}_{\substack{s_0 \sim \iota \\ \mathbf{A}_0 \sim \iota_D^n}} \left[ \sum_{t=0}^T \mathbb{1} \left\{ \bigwedge_{i=1}^n s_0, \ldots, s_{t+1} \models_{L_i} \mathbf{A}_0[i] \right\} \right] \\
&= \mathbb{E}_{\substack{s_0 \sim \iota \\ \mathbf{A}_0 \sim \iota_D^n}} \left[ \sum_{t=0}^T R(s_0, \ldots, s_{t+1}, \mathbf{A}_0) \right],
\end{aligned}
$$

where $s_{t+1} \sim P(s_t, a_t)$, $a_t \sim \pi(s_0, \ldots, s_t, \mathbf{A}_0)$, and $T$ denotes the finite horizon of the game.

For all $(s_0, \ldots, s_t, \mathbf{A}_0) \in S^* \times D^n$, we have $R(s_0, \ldots, s_{t+1}, \mathbf{A}_0) = 1$ if and only if all DFAs in $\mathbf{A}_0$ accept their corresponding labeled traces. We can equivalently write this statement as follows:

$$\mathbf{A}_0/L(s_0), \ldots, L(s_{t+1}) = \mathbf{A}_t/L(s_{t+1}) = \mathbf{A}_\top,$$

which implies that the reward of the Markovian game state $(s_t, \mathbf{A}_t)$ given by Equation (8) is also one, i.e., $R_{\mathcal{D}^n}(\mathbf{A}_t, L(s_{t+1})) = 1$ due to Equations (3) and (4). If $R(s_0, \ldots, s_{t+1}, \mathbf{A}_0) = 0$, then there exists an $i$ such that $\mathbf{A}_0[i]$ doesn't accept the trace labeled with $L_i$, i.e., $\mathbf{A}_t/L(s_{t+1}) \neq \mathbf{A}_\top$, and therefore $R_{\mathcal{D}^n}(\mathbf{A}_t, L(s_{t+1})) = 0$. Thus, rewards for $\pi$ and $\pi'$ are the same with respect to the mapping in Equation (8).

For all $(s_0, \ldots, s_t, \mathbf{A}_0) \in S^* \times D^n$, the probability of transitioning to a next state $(s_0, \ldots, s_{t+1}, \mathbf{A}_0) \in S^* \times D^n$ is given by the undelying Markovian dynamics $P(s_{t+1} \mid s_t, a_t)$ for $a_t \sim \pi(s_0, \ldots, s_t, \mathbf{A}_0)$. For the corresponding Markovian state $(s_t, \mathbf{A}_t)$ given by Equation (8), the probability of transitioning to a next state $(s_{t+1}, \mathbf{A}_{t+1})$ is as follows:

$$P'(s_{t+1}, \mathbf{A}_{t+1} \mid s_t, \mathbf{A}_t) = P(s_{t+1} \mid s_t, a_t)\mathbb{1}\{\mathbf{A}_t/L(s_{t+1}) = \mathbf{A}_{t+1}\},$$

i.e., transition using the same underlying Markovian dynamics $P(s_{t+1} \mid s_t, a_t)$ and mask based on the deterministic product DFA transition $\mathbb{1}\{\mathbf{A}_t/L(s_{t+1}) = \mathbf{A}_{t+1}\}$. Since $(s_0, \ldots, s_t, \mathbf{A}_0) \mapsto (s_t, \mathbf{A}_t)$ and any next state $(s_0, \ldots, s_{t+1}, \mathbf{A}_0) \mapsto (s_{t+1}, \mathbf{A}_{t+1})$, we have that $\mathbb{1}\{\mathbf{A}_t/L(s_{t+1}) = \mathbf{A}_{t+1}\} = 1$ by the construction of the mapping given in Equation (8). Therefore, for any transition from $(s_0, \ldots, s_t, \mathbf{A}_0)$ to $(s_0, \ldots, s_{t+1}, \mathbf{A}_0)$ in the non-Markovian game, there exists a transition (given by Equation (8)) with the same probability from $(s_t, \mathbf{A}_t)$ to $(s_{t+1}, \mathbf{A}_{t+1})$ in the Markovian reformulation of the game. Hence, the transitions for $\pi$ and $\pi'$ are equivalent with respect to deterministic mapping in Equation (8).

Finally, as the rewards and transition probabilities for $\pi$ and $\pi'$ are the same under the mapping given in Equation (8), the optimal values for $J(\pi)$ from Equation (1) and $J'_\gamma(\pi')$ from Equation (5) are the same as $\gamma \to 1^-$, which completes the proof. $\square$

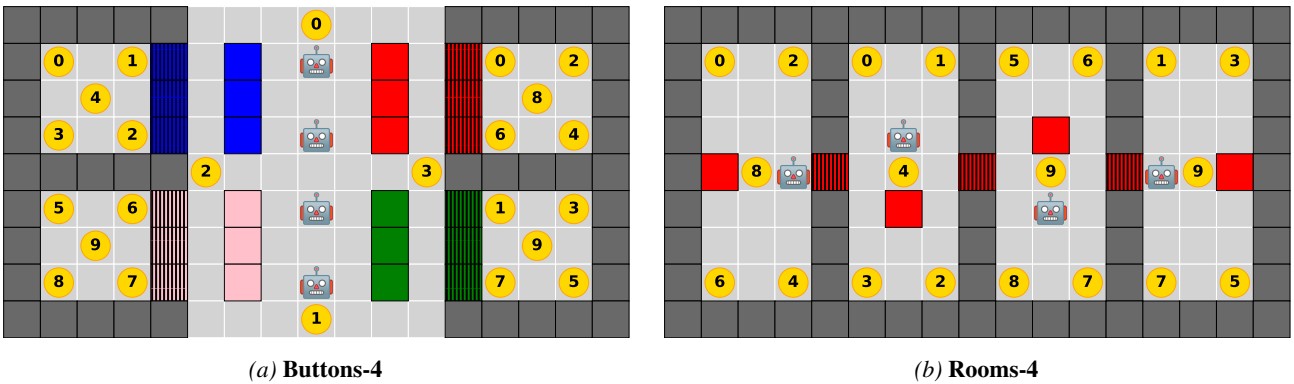

*(a)* **Buttons-4**                                                    *(b)* **Rooms-4**

*Figure 6.* Considered four-agent variations of `TokenEnv`.

## B. Implementation Details

In this section, we discuss the details of our practical Python implementation[2] of the theoretical framework given in Section 3. We follow the practices presented by PureJaxRL (Lu et al., 2022) and JaxMARL (Rutherford et al., 2024) and implement our framework in JAX (Bradbury et al., 2018), an accelerator-oriented automatic differentiation library.

### B.1. Environments

We introduce a new environment called `TokenEnv`[3], a fully observable discrete multi-agent environment implemented in JAX. Agents observe the global state from their own point of view, i.e., one-hot encodings of objects appear relative to the agent's position, and synchronously move in four cardinal directions or stay where they are. There are different tokens in the environment. If an agent is on a token, then its state is labeled as that token, i.e., *agents do not collect tokens but reach tokens*, and therefore tokens are unlimited. At the beginning of every episode, each agent is assigned an a priori unknown DFA task defined over these tokens.

We consider two- and four-agent variants of `TokenEnv` and four layouts in total: **Buttons-2** and **Buttons-4** given in Figures 1 and 6a, respectively, and **Rooms-2** and **Rooms-4** given in Figures 2 and 6b, respectively. In all layouts, to go through the striped colored cells—serving as closed doors, an agent needs to be on the corresponding colored cells—serving as buttons. Therefore, agents must cooperate to complete their tasks. **Buttons-2** and **Buttons-4** require agents to reason about other agents' DFA tasks to press the correct buttons; and **Rooms-2** and **Rooms-4** emphasize asymmetric conditions agents can be in, therefore providing a testbed for evaluating the impact of optimal task assignments.

### B.2. DFA Distributions

We implement a native JAX package, called `DFAx`[4], facilitating all operations on DFAs defined in this paper. For DFA minimization, we implement a parallel algorithm called *naive partition refinement* as described in (Martens & Wijs, 2024). `DFAx` package includes samplers for generating different types of DFA tasks: (i) `Reach` DFAs order tokens, e.g., "reach 1 and then reach 2," (ii) `ReachAvoid` DFAs order tokens with hard constraints, i.e., unrecoverable conditions, e.g., "reach 1 while avoiding 2," and (iii) `ReachAvoidDerived` (RAD) DFAs are randomly mutated `Reach` and `ReachAvoid` DFAs, e.g., "reach 1 while avoiding 2, and if you reach 3 before reaching 1, then you must reach 4 too", and therefore RAD DFAs define a richer task structure than `Reach` and `ReachAvoid`. Random algorithms for sampling these DFAs are given in Algorithms 1 to 3, and example DFAs sampled from these distributions are given in Figures 7a to 7c, respectively.

RAD DFAs were first introduced in (Yalcinkaya et al., 2024) as a prior distribution for learning automata-conditioned policies in the single-agent setting. It was shown that policies trained on RAD DFAs can generalize to other DFA classes such as `Reach` and `ReachAvoid`. Thus, to learn agents that can handle a large class of tasks, we use the RAD DFA distribution as our prior. During training, we sample RAD DFAs with at most 5 states, where the number of states is sampled uniformly. At

---

[2]Available at https://github.com/rad-dfa/acc-marl.
[3]Available at https://github.com/rad-dfa/dfa-gym.
[4]Available at https://github.com/rad-dfa/dfax.

the beginning of each episode, for $n$ agents, we sample $n$ `RAD` DFAs such that at least one of these DFAs is non-trivial, i.e., not trivially accepting or rejecting. However, we still allow sampling of trivially accepting DFAs to instantiate *helper agents*, i.e., agents that are not assigned a DFA but are there to help others. Algorithm 4 presents our sampling procedure for assigning random DFA tasks (from a given distribution) to agents in a Markov game.

To facilitate the proposed solution for mitigating history dependency, we augment the underlying environment, i.e., `TokenEnv`, and include agents' latest minimal DFA tasks in returned observations, as described in Section 3.2. For the credit assignment problem, we return the shaped reward defined in Section 3.3. Finally, to learn a provably correct DFA encoder addressing the representation bottleneck challenge, we train a GATv2 (Brody et al., 2022) over `RAD` DFAs as described in (Yalcinkaya et al., 2025) with a few minor changes discussed below.

### B.3. Pretraining RAD Embeddings

We pretrain a GATv2 encoder over the induced DFA space of `RAD` DFAs as defined in Equation (2). During this pretraining, we sample `RAD` DFAs with at most 10 states sampled from a geometric distribution. As we use JAX for our practical implementation, which enforces new constraints over the code, such as working with fixed-sized arrays, we introduce changes to the DFA featurization and the GATv2 architecture used in (Yalcinkaya et al., 2025), listed below, respectively.

1. In both (Yalcinkaya et al., 2024) and (Yalcinkaya et al., 2025), DFA featurization involves reinterpreting transitions of a DFA as nodes and encoding constraints of these transitions as one-hot node features, which substantially increases the number of nodes in DFA featurization. Instead, we interpret the DFA transitions as edges and encode the constraints as one-hot edge features.
2. Consequently, we adapt our GATv2 architecture to accommodate edge features rather than solely using node features.

Given a DFA $\mathcal{A} = \langle Q, \Sigma, \delta, q_0, F \rangle$, we construct its featurization $G = (V, E, h, e)$, representing as a graph, where

- $V$ is the nodes, containing a node for each state of $\mathcal{A}$,
- $E$ is the edges, containing an edge for each transition of $\mathcal{A}$,
- $h$ is the node features, i.e., one-hot vectors encoding whether states are initial, accepting, rejecting, or neither,
- $e$ is the edge features, i.e., one-hot vectors encoding constraints of their corresponding transitions.

We refer to the features of a node $v \in V$ as $h_v$ and the features of an edge between nodes $v, u \in V$ as $e_{vu}$. In our GATv2 implementation, at each message passing step, node features are updated as:

$$h'_v = \sum_{u \in N^{-1}(v)} \alpha_{vu} W_{msg} \left[ h_u \parallel e_{vu} \right],$$

where $N^{-1}(v)$ is the set of nodes with edges to $v$, $W_{msg}$ is a linear map, and $\alpha_{vu}$ is the attention score between $v$ and $u$ computed as:

$$\alpha_{vu} = \mathrm{softmax}_v \left( a^\top \mathrm{LeakyReLU} \left( W_{atn} \left[ h_v \parallel e_{vu} \parallel h_u \right] \right) \right),$$

where $a$ is a vector and $W_{atn}$ is a linear map. For a DFA with $n$ states, we perform $n$ message passing steps, which guarantees that the node representing the initial state of the DFA has received messages from all $n$ nodes. Therefore, we pick the feature vector of this node as the embedding of the corresponding DFA task $\mathcal{A}$. All other details are identical to those presented in (Yalcinkaya et al., 2024; 2025).

### B.4. Decentralized Policies

We learn a single policy deployed for each agent independently. The policy architecture for an agent $i$ is given in Figure 4. An agent $i$, respectively, takes its agent ID $i$, the global state from its own point of view, denoted by $s_i$, its assigned DFA task $\mathcal{A}_i$, and the DFAs of other agents ordered by their IDs. We pass the DFAs through the pretrained (and frozen) DFA encoder and use these DFA embeddings along with the ID and state embeddings to compute a task embedding, which is a latent representation of the current task of the agent, potentially encoding information about whether to help another agent or work on its own task. This task embedding, along with the ID and state embeddings, is then used to compute the agent's next action. We also use a critic (not shown in Figure 4) predicting state values and use this value function for the optimal assignment of tasks to agents as described in Section 3.5. Below, we present the details of the policy architecture in Figure 4.

- **Embed Layer** maps agent IDs to 32-dimensional vectors.
- **CNN** is a convolutional neural network with layers [16, 32, 64], each with a 2x2 kernel and ReLU activation.

- **Task Encoder** is a multilayer perceptron with layers [256, 256, 32], each with a tanh activation function.
- **Policy** is a multilayer perceptron with layers [64, 64, 64, 5], each with a ReLU activation, where 5 is the number of actions, i.e., four cardinal directions and a noop action. It outputs a categorical distribution over the actions, and to get an action, we sample from this distribution.
- **Value**, not presented in Figure 4, is a multilayer perceptron with layers [64, 64, 1], each with a ReLU activation.

We train using Independent Proximal Policy Optimization (IPPO) (de Witt et al., 2020; Lu et al., 2022; Rutherford et al., 2024)—the hyperparameters used for **Buttons-2** and **Rooms-2** environments are given in Table 2. We use the same hyperparameters for **Buttons-4** and **Rooms-4**, except that we set the entropy coefficient to $0.05$ to further encourage exploration. Finally, we note that in **Buttons-4** and **Rooms-4**, for policies without pretrained RAD Embeddings, we had to limit the number of environments to 32 because otherwise the GPU memory got exhausted trying to fit all parameters, which highlights the use of RAD Embeddings as a means to reduce the number of learned parameters.

| Hyperparameter | Value |
| --- | --- |
| Learning rate | $3 \times 10^{-4}$ |
| Number of environments | 64 |
| Number of steps per rollout | 1024 |
| Total timesteps | 10,000,000 |
| Update epochs | 8 |
| Number of minibatches | 8 |
| Discount factor ($\gamma$) | 0.99 |
| GAE $\lambda$ | 0.95 |
| Clipping coefficient | 0.2 |
| Entropy coefficient | 0.02 |
| Entropy coefficient decay | False |
| Value function coefficient | 0.5 |
| Max gradient norm | 0.5 |

*Table 2.* IPPO hyperparameters used in experiments.

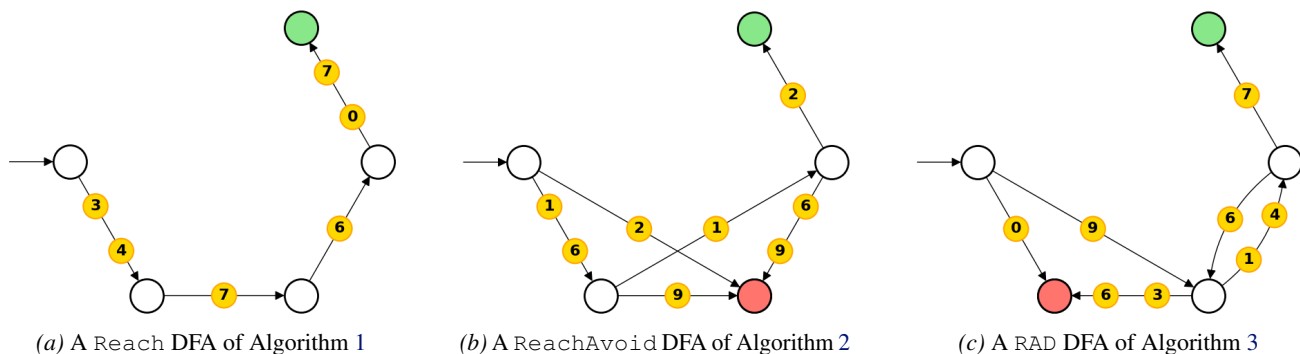

*(a)* A `Reach` DFA of Algorithm 1      *(b)* A `ReachAvoid` DFA of Algorithm 2      *(c)* A `RAD` DFA of Algorithm 3

*Figure 7.* Different types of DFA tasks sampled from the task distributions considered in the paper.

---

**Algorithm 1** `Reach` DFA Sampler

---

1: Sample a sequence one-step `Reach` problems of length $k - 1$ as a DFA, where $k \sim$ Uniform, call it $\mathcal{A}$.
2: For each stuttering symbol, i.e., a symbol that does not change the state, of $\mathcal{A}$, make it `Reach` with 0.1 probability; otherwise keep it unchanged.
3: **return** $\min(\mathcal{A})$

---

**Algorithm 2** `ReachAvoid` DFA Sampler

---

1: Sample a sequence one-step `ReachAvoid` problems of length $k - 1$ as a DFA, where $k \sim$ Uniform, call it $\mathcal{A}$.
2: For each stuttering symbol, i.e., a symbol that does not change the state, of $\mathcal{A}$, make it `Reach` or `Avoid` with 0.1 probability; otherwise keep it unchanged.
3: **return** $\min(\mathcal{A})$

---

**Algorithm 3** `ReachAvoidDerived` (`RAD`) DFA Sampler

---

1: Sample a sequence of one-step `Reach` and `ReachAvoid` problems of length $k - 1$ as a DFA, where $k \sim$ Uniform and at each step, flip a coin to decide whether to include the `Avoid` part, call it $\mathcal{A}$.
2: For each stuttering symbol, i.e., a symbol that does not change the state, of $\mathcal{A}$, make it `Reach` or `Avoid` with 0.1 probability; otherwise keep it unchanged.
3: $\mathcal{A} \leftarrow \min(\mathcal{A})$
4: **for** $i = 1$ to $m$ (where $m \sim$ Uniform) **do**
5:     $\mathcal{A}' \leftarrow$ Mutate $\mathcal{A}$, i.e., randomly change a transition
6:     $\mathcal{A}' \leftarrow$ Make accepting states of $\mathcal{A}'$ sinks
7:     $\mathcal{A}' \leftarrow \min(\mathcal{A}')$
8:     **if** $\mathcal{A}'$ is not a trivial DFA (i.e., $\mathcal{A}_\top$ or $\mathcal{A}_\bot$) **then**
9:         $\mathcal{A} \leftarrow \mathcal{A}'$
10:     **end if**
11: **end for**
12: **return** $\mathcal{A}$

---

**Algorithm 4** Multi-Agent DFA Sampler

---

1: Sample $n_{\text{trivial}} \sim \text{Uniform}[0, n)$, where $n$ is the number of agents in the multi-agent environment.
2: Generate $n_{\text{trivial}}$ many accepting DFAs, call it $\mathbf{A}_{\text{trivial}}$.
3: Sample $n - n_{\text{trivial}}$ many DFAs form $\iota_D$, where $\iota_D$ is the given DFA distribution, call it $\mathbf{A}_{\text{non-trivial}}$.
4: $\mathbf{A} \leftarrow$ Concatenate $\mathbf{A}_{\text{trivial}}$ and $\mathbf{A}_{\text{non-trivial}}$
5: $\mathbf{A} \leftarrow$ Shuffle $\mathbf{A}$
6: **return** $\mathbf{A}$

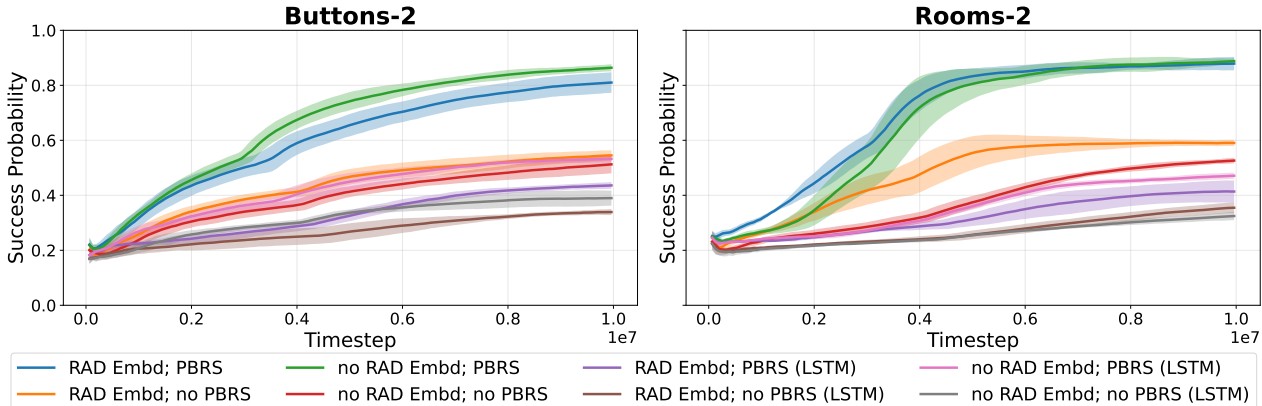

*Figure 8.* Success probabilities of learned policies throughout training, reported over 5 random seeds—shaded regions indicate standard deviation. "RAD Embd; PBRS" refers to Markovian policies conditioning on pretrained RAD Embeddings and trained with the shaped reward, i.e., the full solution proposed in Section 3, and "(LSTM)" suffix denotes history-dependent policies.

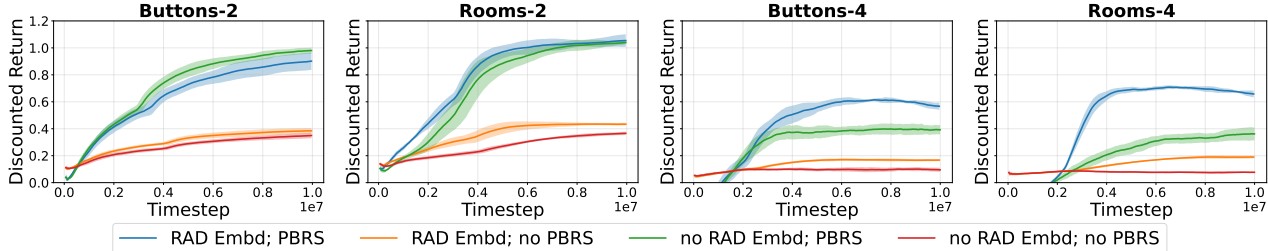

*Figure 9.* Discounted returns of learned policies throughout training, reported over 5 random seeds—shaded regions indicate standard deviation. "RAD Embd; PBRS" refers to Markovian policies conditioning on pretrained RAD Embeddings and trained with the shaped reward, i.e., the full solution proposed in Section 3. Results with the history-dependent baseline are in Figure 10.

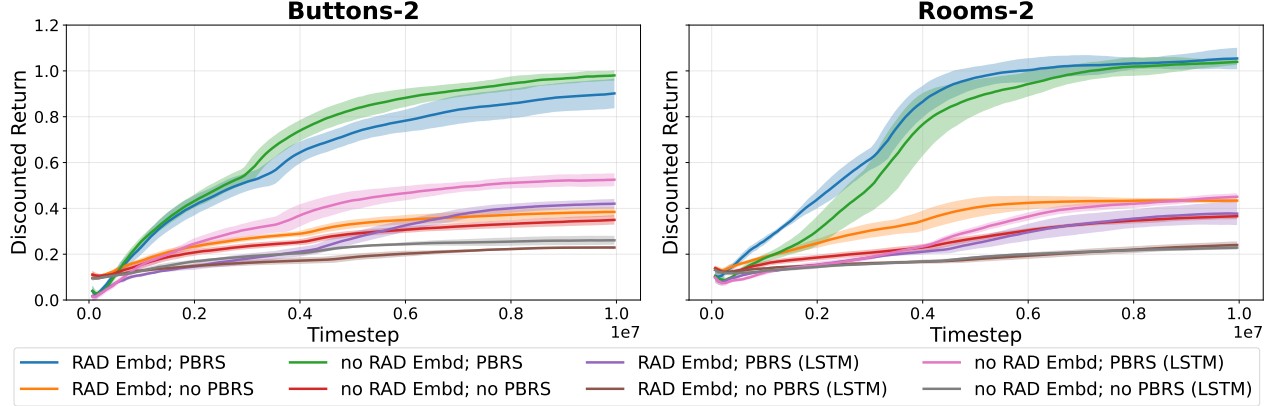

*Figure 10.* Discounted returns of learned policies throughout training, reported over 5 random seeds—shaded regions indicate standard deviation. "RAD Embd; PBRS" refers to Markovian policies conditioning on pretrained RAD Embeddings and trained with the shaped reward, i.e., the full solution proposed in Section 3, and "(LSTM)" suffix denotes history-dependent policies.

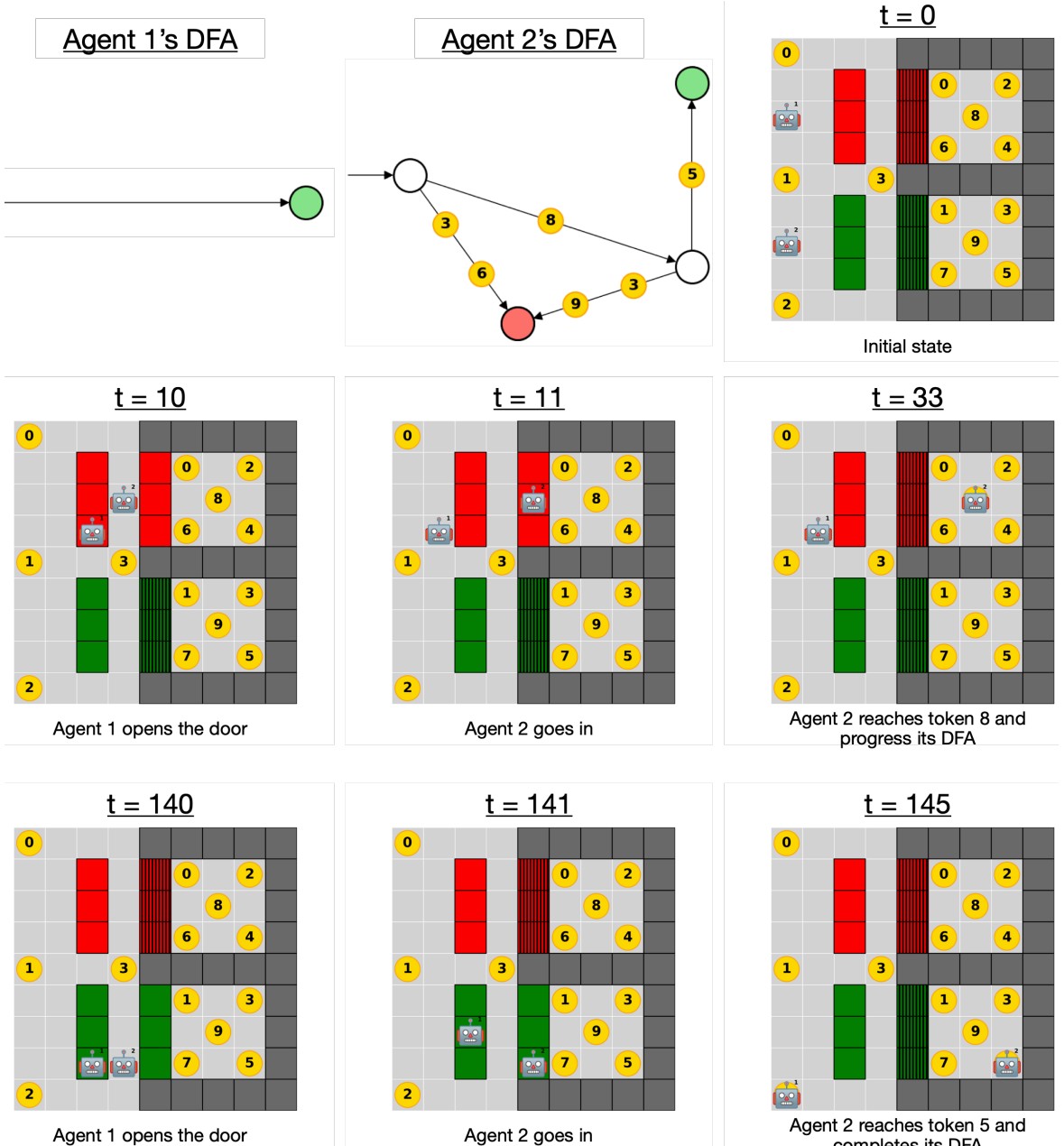

*Figure 11.* Agent 1 in the helper agent role, assisting agent 2 throughout the episode.

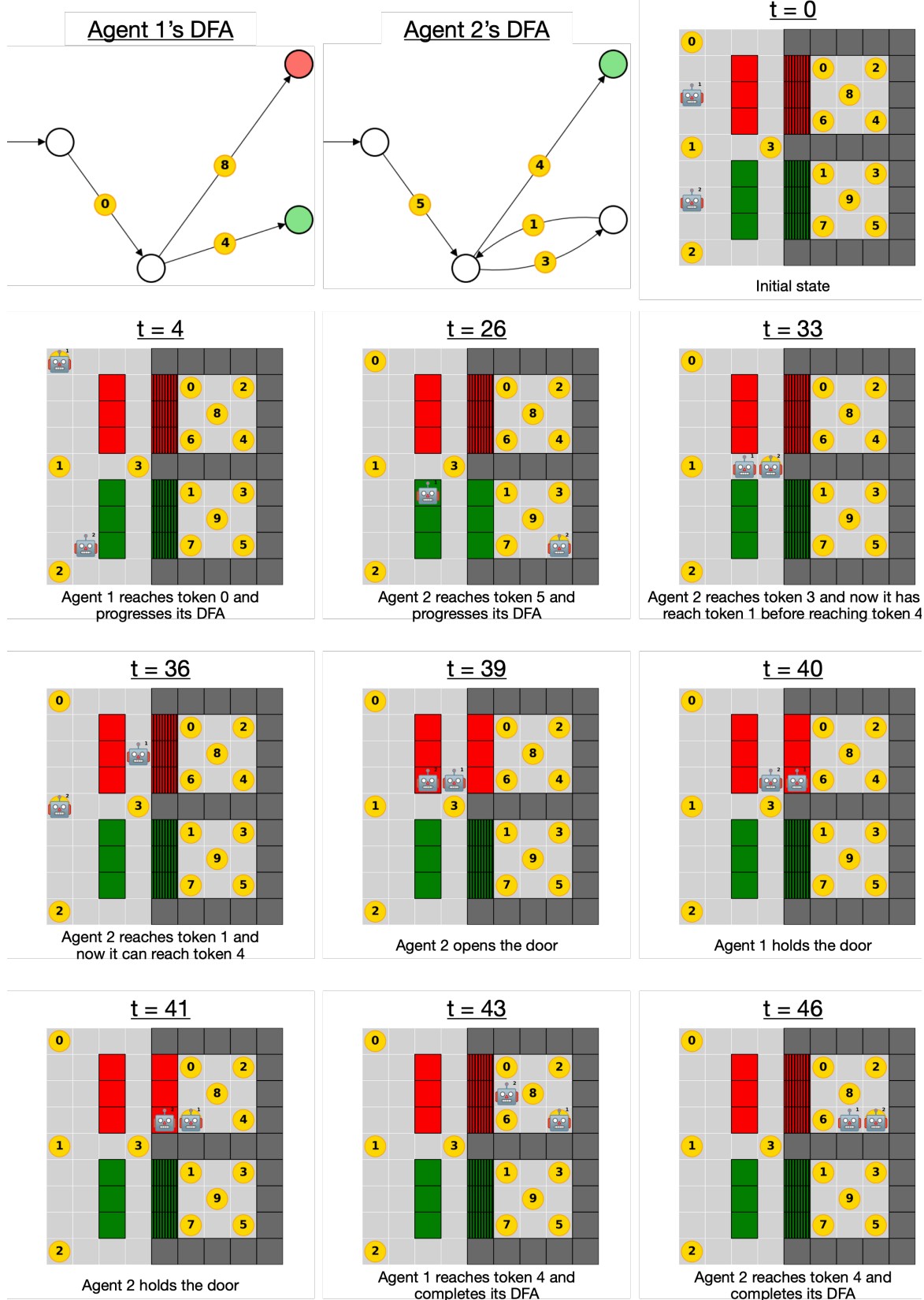

*Figure 12.* Agents holding the door for each other and completing both tasks.

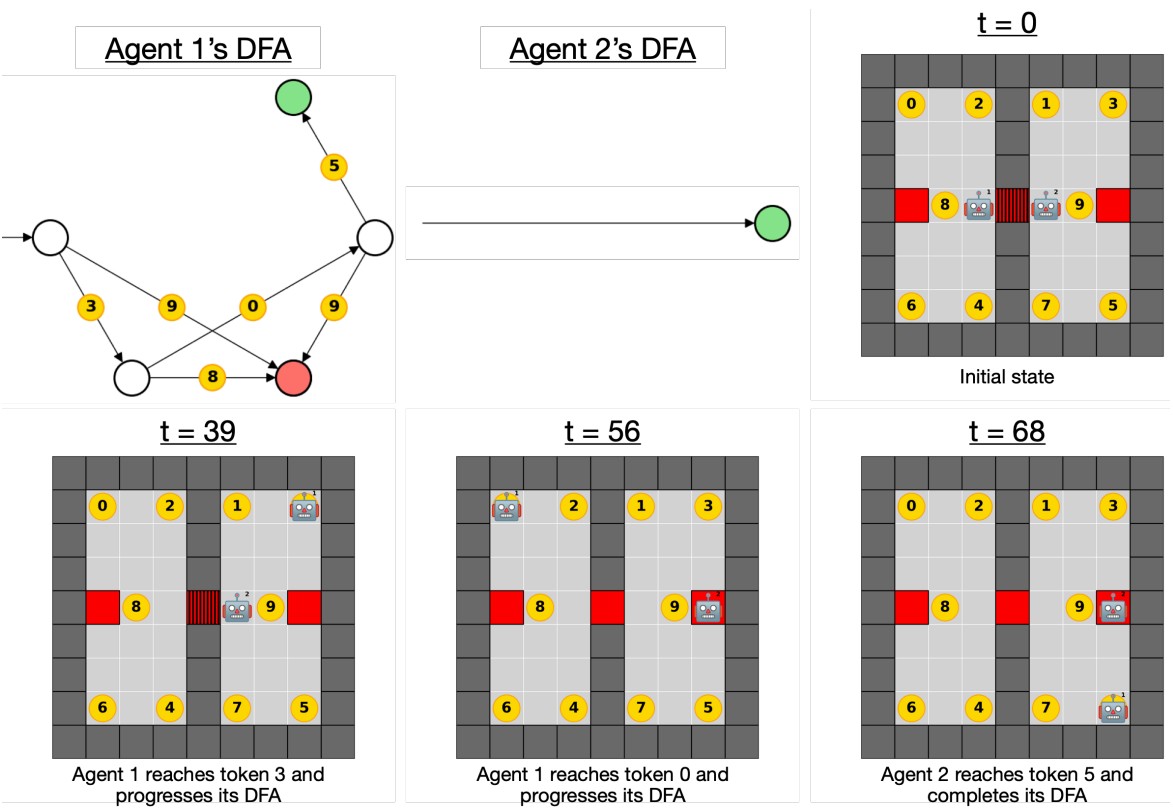

*Figure 13.* Agent 2 in the helper agent role, assisting agent 1 throughout the episode.

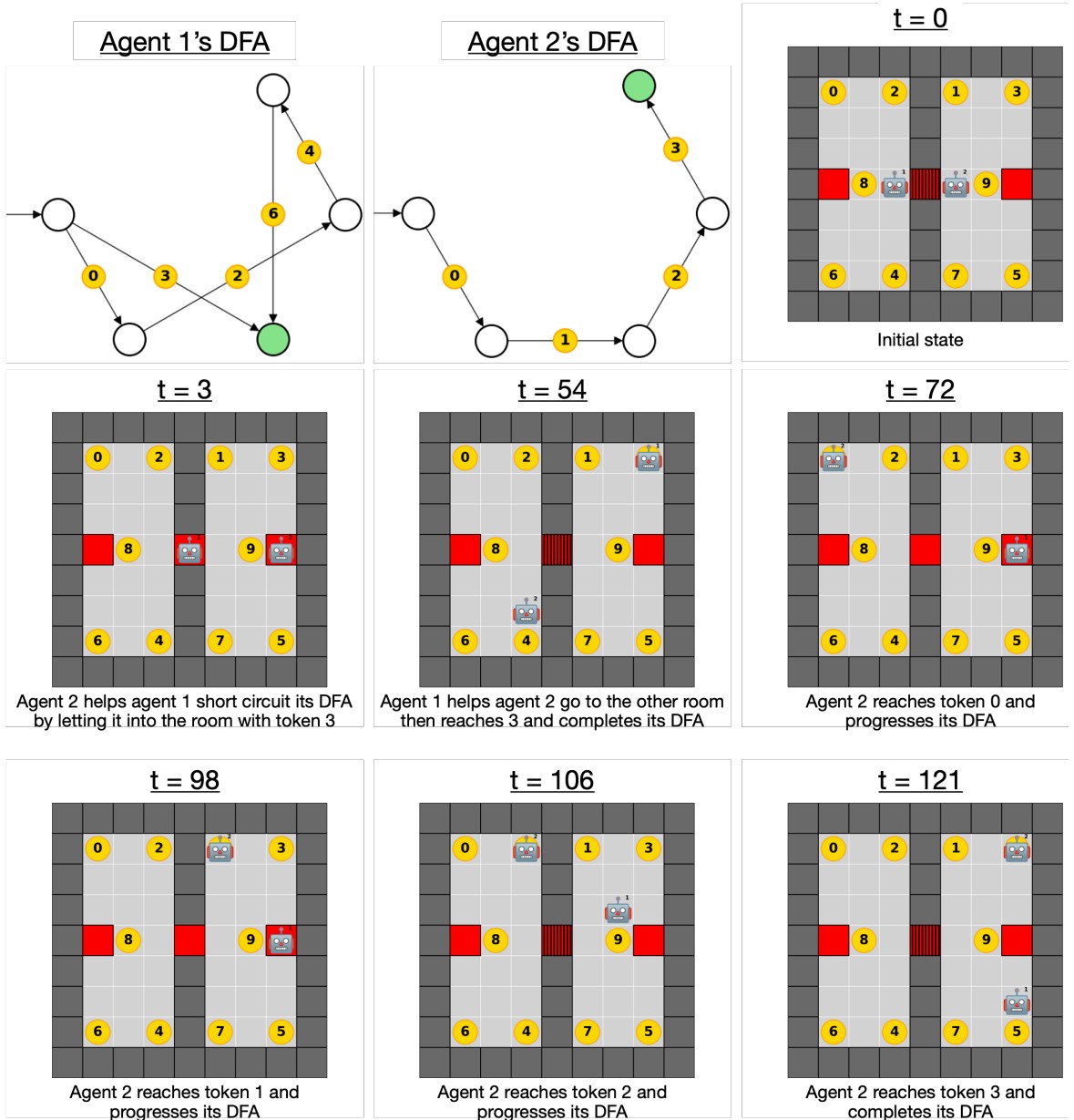

*Figure 14.* Agents picking the shortest path in agent 1's DFA and completing both tasks.

