# OpenReview forum: "Automata-Conditioned Cooperative Multi-Agent Reinforcement Learning"
_ICML.cc/2026/Conference — ICML 2026 regular_

### Official Review · Reviewer_ArTf · 2026-03-06

**Soundness:** 4
**Presentation:** 4
**Significance:** 3
**Originality:** 3
**Overall Recommendation:** 5
**Confidence:** 4

**Summary:**

Each episode, a set of $n$ agents gets a new set of $n$ tasks that need to be performed; each of these tasks are given in the form of a DFA, where each transition represents completion of a subtask. ACC-MARL solves two sub-problems: what is the best way to assign these tasks/DFAs to each agent, and given these assignments, how can agents complete their tasks?

The latter sub-problem is solved by learning individual policies where in addition to the environment observation, the policies are conditioned on an embedding of all agents' DFAs and the progress so far through each of them (with a nicer implementation; the embedding is actually of a quotient DFA rather than of the original DFA plus the current state). Agents receive a reward when all tasks are complete, with some potential-based reward shaping to help each agent complete individual tasks.

The value functions from these learned individual policies can be used to test if a given agent is likely to be able to complete a given task/DFA. At the beginning of an evaluation episode, ACC-MARL tests all permutations of agent-DFA assignments, and selects an assignment on the Pareto frontier of value estimates.

**Compliance With Llm Reviewing Policy:**

Affirmed.

**Final Justification:**

Rebuttal did not change original assessment.

**Key Questions For Authors:**

Q1. Not really a question that needs to be answered but more of a comment/suggestion based on the no-RAD configurations eventually surpassing the RAD configurations in the 2-agent environments. Did you run any trials where the embeddings are pre-trained according to Yalcinkaya et al. 2025, but unfrozen to be fine-tuned (potentially with a warm-up phase before unfreezing the weights)?

**Limitations:**

Yes

**Strengths And Weaknesses:**

S1. The proposed method provides a sound and experimentally effective method for specifying tasks for multiple agents to complete, and optimally assigning each agent to a task. The presentation is clear enough that I think I could re-implement the method based on the text alone.

S2. The authors promise to release several Python packages that, based on their description, will be useful to researchers even for somewhat unrelated projects.

W1. The novelty is primarily in the multi-agent task assignment (third paragraph of summary); learning the individual policies (second paragraph of summary) are similar to existing specification-guided RL literature (most of which are listed in first paragraph of Section 6).

W2. Some elements of the problem setting seem unlikely; for example, that one agent---and we don't care which one---may need to reach area 1 while avoiding area 2, but other agents are allowed to enter area 2 as much as they please. This may be more of a quirk of the specific evaluation environment and DFA distribution, rather than a limitation of ACC-MARL itself.

---

> ### Author Rebuttal · Authors · 2026-03-27
>
> We thank the reviewer for their thoughtful and constructive feedback. We address each point below.
>
> # Novelty (W1)
> We agree that certain components of our framework build on prior specification-guided RL works. Our primary contribution is extending these ideas to a multi-task MARL setting, where a team of agents is assigned a set of DFAs sampled at the beginning of each episode. To our knowledge, ACC-MARL is the first to use formal specifications in a multi-task MARL setting, as prior work in this domain is either limited to multi-task RL or single-task MARL, but cannot handle multi-task MARL. We agree that the optimal task assignment component is itself another key novelty of our work. We will make these details more explicit in the final version.
>
> # Problem setting (W2)
> We thank the reviewer for pointing out this detail. Our framework naturally supports global constraints of the type described by the reviewer. For example, if all agents must avoid token 2, this can be encoded by adding an “avoid” transition on token 2 to each agent’s DFA. More generally, while each DFA defines a local task for an agent, global “avoid” constraints can be incorporated by including them in all agents’ specifications. As such, the setting considered in the paper allows both local and global “avoid” constraints. We will clarify this in the final version.
>
> # Fine-tuning RAD Embeddings (Q1)
> We explored fine-tuning RAD embeddings in preliminary experiments and did not observe improved performance. This is consistent with findings of the original work on RAD embeddings (Yalcinkaya et al., 2024), where the authors report that keeping the pretrained DFA encoder frozen yields better performance than fine-tuning, as freezing preserves the structure induced by bisimulation-based pretraining. We will include a brief discussion of this in the final version.

---

> > ### Author Rebuttal · Reviewer_ArTf · 2026-03-31
> >
> > Acknowledged.

---

### Official Review · Reviewer_1mv2 · 2026-03-11

**Soundness:** 3
**Presentation:** 4
**Significance:** 3
**Originality:** 4
**Overall Recommendation:** 5
**Confidence:** 4

**Summary:**

This paper studies cooperative MARL under the centralized training and decentralized execution setting. Each agent is assigned a temporal task represented by a DFA at runtime. A task is considered complete only when all its subgoals are achieved. To address the history-dependence problem, the authors update each agent’s DFA online as the episode unfolds. During MARL training, the method uses sub-DFAs to provide dense rewards while still preserving the optimality of the overall task, which helps mitigate the sparse reward problem. To make DFAs more suitable for neural networks and to improve generalization, the paper encodes the DFA using a GNN and uses the resulting embedding to represent the task. The ablation studies clearly demonstrate the effectiveness of the proposed method.

**Compliance With Llm Reviewing Policy:**

Affirmed.

**Final Justification:**

I recommend acceptance of this paper.

**Key Questions For Authors:**

Please address my concerns in Strengths And Weaknesses.

**Limitations:**

The evaluation is restricted to a custom, fully observable, discrete environment with only two- and four-agent teams, so it is hard to tell whether the proposed framework would remain effective in more realistic MARL settings with partial observability, noisy perception, learned label abstractions, or substantially larger teams. In addition, the task-assignment procedure appears to rely on enumerating task permutations, which raises an obvious scalability concern. Finally, the method depends on a pretrained frozen automata encoder, so some of the gains may be tied to representation pretraining rather than the MARL formulation alone. These issues do not invalidate the contribution, but they do limit how broadly the current results can be interpreted.

**Strengths And Weaknesses:**

### Pros
1. The idea is novel and problem definition is clear. The paper separates history dependence, credit assignment, and representation bottleneck cleanly, and each fix is connected back to an optimality argument rather than being presented as a heuristic patch.
2. The ablations proves its effectiveness. The experiments show that reward shaping is essential, that history-dependent LSTM variants struggle, and that pretrained RAD embeddings become more useful as team size and coordination difficulty increase.
3. In asymmetric layouts, optimal assignment based on learned values improves success, which makes the framework more practically interesting than a pure policy-learning paper.

### Cons
1. Task assignment may not scale. The assignment rule explicitly optimizes over perm(A), so the obvious concern is factorial growth with team size; the paper shows the idea in small cases but not at larger scale.
2. The evaluation is strongest as an internal ablation study. I would have liked broader head-to-head comparisons against external formal-spec MARL baselines (e.g. LTL) from the related-work set, because that would clarify how much of the gain comes from the new multi-agent formulation versus the inherited automata machinery.

---

> ### Author Rebuttal · Authors · 2026-03-27
>
> We thank the reviewer for their thoughtful and constructive feedback. We address each point below.
>
> # Task assignment
> We agree with the reviewer that the approach presented for optimal task assignment may not scale to larger teams because it enumerates all assignments. However, we emphasize that this step is performed once per episode and is orthogonal to the other contributions of the paper.
>
> The assignment problem itself is a standard combinatorial optimization problem, and scalable approximations (e.g., greedy or learned assignment policies) can be directly substituted without modifying the rest of the framework. As such, our approach provides an exact reference solution for evaluating approximate methods for the optimal task assignment problem in MARL, which is a promising direction for future work that needs further in-depth investigation.
>
> We will include this discussion in the final version using the extra page.
>
> # Baselines
> We agree that comparisons to external formal-specification-guided MARL methods would strengthen the evaluation. However, existing approaches are designed for single-task MARL (Neary et al., 2021; Smith et al., 2023; Shah et al., 2025a), whereas our setting is multi-task, i.e., tasks are sampled and assigned at the beginning of each episode. This mismatch makes direct comparisons non-trivial, as these methods do not support dynamic multi-task assignment, i.e., they train MARL policies for a fixed, single task and thus have to be retrained for new DFAs assigned at runtime. As such, to the best of our knowledge, ACC-MARL is the first to use formal specifications in a multi-task MARL setting. Therefore, we focus on ablations that isolate the contributions of each component of our solution.
>
> We will clarify this distinction in the final version.
>
> # Limitations
> We thank the reviewer for articulating these limitations. We think that scaling to substantially larger teams and more realistic settings is a promising direction for future work. As pointed out by the reviewer, the use of RAD Embeddings to represent DFAs is an important component of our method, enabling ACC-MARL to scale to 4-agent teams while maintaining a similar sample efficiency to the 2-agent case. We note that our work is the first to use RAD Embeddings in MARL and to show that they improve sample efficiency in learning decentralized team policies.
>
> In the final version, we will use the additional page to expand the discussion of future work and the limitations of our framework.

---

> > ### Author Rebuttal · Reviewer_1mv2 · 2026-04-03
> >
> > My questions have been fully addressed, and I have increased my score. As a limitations section is required by International Conference on Machine Learning, I encourage the authors to include one in the revision.

---

### Official Review · Reviewer_874m · 2026-03-12

**Soundness:** 3
**Presentation:** 2
**Significance:** 3
**Originality:** 3
**Overall Recommendation:** 4
**Confidence:** 4

**Summary:**

The authors propose ACC-MARL, a framework for MARL under DFA task encodings. The authors use a centralized training-decentralized execution approach to predict its next best action given its own DFA, as well as the DFA of other agents. They propose a minimal product DFA representation to determinize the problem, i.e., render it Markovian. Reward shaping provides a denser signal that helps agents measure their own progress towards satisfaction. DFAs are embedded using a RAD framework to help manage bottlenecks and encourage generalizability. Results are demonstrated in two toy domains.

**Compliance With Llm Reviewing Policy:**

Affirmed.

**Final Justification:**

Overall, I think this is a good paper, and I maintain my overall positive assessment.

**Key Questions For Authors:**

What is the cost of determinization and the other steps in 3.2. In an automata theoretic context, these are usually very expensive. Can the authors elaborate on the benefits and drawbacks to this approach?

Is my understanding correct that assignment is over the entire DFA 3.5? A “task” in this sense is a DFA? That is, DFAs are changed or combined.

**Limitations:**

Yes

**Strengths And Weaknesses:**

Overall, this paper has good ideas and results. The writing and organization could use a bit of improvement, but are ok for the most part. Some of the authors conclusions appear a bit strong compared to the evidence they present.

There is an implicit assumption for most of the paper that each agent has its own DFA. This is fine, but it would be good if the authors were very clear about this. It wasn’t obvious to me until I read the task assignment piece.

The conclusions are a bit strong based on the evidence. For example, A2 says that policies “seamlessly” scale. This is not very objective language. What does it mean to be seamless or not? I recommend removing it. Further, scaling from 2 to 4 is a nice result, but if this is the size limit, it’s not a very exciting result in terms of scalability. Perhaps “generalizing” to 4 agents is a better term.

I find the generalization argument similarly underwhelming. Looking at the tasks, it seems that Reach and ReachAvoid tasks are essentially simple cases of the RAD DFAs considered. Is that true? In that sense, aren’t the OOD results the ones that are worth highlighting? In that setting, I’d appreciate clarity on how they are OOD. Is it only size? What about transitions? Task type? Propositions?

Minor:
-	Gamma is used in 2.1 but not defined there. I assume this is the discount factor.

---

> ### Author Rebuttal · Authors · 2026-03-27
>
> We thank the reviewer for their thoughtful and constructive feedback. We address each point below.
>
> # Cost Section 3.2 and benefits/drawbacks of DFAs
> In practice, the DFA space formulation given in Equation 2 is never explicitly computed or stored in memory; instead, DFA spaces are implicitly defined by generating DFA distributions, e.g., RAD DFAs &mdash; see Algorithm 3 in the Appendix of the paper for how RAD DFAs are sampled. For each agent, at the beginning of an episode, we sample a DFA, and then, at every step, we progress the DFA based on the observed alphabet symbol.
>
> DFA progression involves updating the current DFA state using the observed symbol and then minimizing the resulting DFA, as defined in Section 2. The former step has $\mathcal{O}(1)$ complexity, i.e., just read the next state from the transition table of the DFA. For a DFA with $n$ states, the computational cost of DFA minimization is $\mathcal{O}(n\log n)$ when one uses Hopcroft’s algorithm, which is the best-known bound. Therefore, the cost of the steps presented in Section 3.2 is negligible.
>
> DFAs provide several major benefits to our framework. First, their well-defined operational semantics enable efficient DFA progression as described above. Second, the compositional nature of DFAs enables a complex task to be decomposed into simpler sub-tasks that can be assigned to individual agents, while still ensuring satisfaction of the overall specification (Neary et al., 2021; Smith et al., 2023; Shah et al., 2025a). Third, DFAs are expressive for finite-horizon behaviors: any finite behavior over an alphabet can be represented as a path in a DFA, and a set of such behaviors can be specified by encoding each finite behavior as a path and then minimizing the resulting DFA to get a compact task representation. Hence, the use of DFAs is crucial to the scalability and expressivity of our framework.
>
> The main drawback of using DFAs is the assumption of a fixed labeling function per agent, which, as defined in Problem 3.1, maps environment observations to alphabet symbols and thus defines the semantics of DFA tasks. In our current setting, these labeling functions are known and remain fixed across training and evaluation. Extending our framework to support dynamic or learned labeling functions, i.e., an open or evolving alphabet where task semantics may change across episodes, is an important direction for future work, but it introduces additional challenges that warrant in-depth investigation in their own right and are beyond the scope of this paper.
>
> We plan to include the discussed details in the final version using the provided extra page.
>
> # Task assignment (Section 3.5)
> Yes, each task corresponds to a DFA, and the team is given a set of such tasks. Throughout the paper, “task” is used interchangeably with “DFA” as we consider DFAs to be a particular modality specifying tasks. Section 3.5 presents a method for assigning a set of DFAs to a multi-agent team by optimizing over all possible task assignments using the proxy value function defined in the paper. We will clarify this terminology in the final version to avoid ambiguity.
>
> # Scalability
> We agree that the current evaluation, scaling from 2 to 4 agents, focuses on a relatively small number of agents, and we will revise the wording in the paper accordingly.
>
> The scale of our experiments reflects the evaluation scope rather than a limitation of the framework. ACC-MARL follows a centralized training, decentralized execution paradigm, where execution is per-agent and does not require joint action enumeration. As a result, for each agent, the DFA component of the game state scales linearly with the number of agents, i.e., in a team of $n$ agents, each agent conditions on $n$ DFAs. We chose scaling from 2 to 4 agents as a controlled experiment to study the effect of doubling the number of agents on learning optimal ACC-MARL policies. The results show that our approach generalizes to 4 agents while maintaining similar sample efficiency to the 2-agent setting.
>
> Applying ACC-MARL in larger agent populations, especially in the context of swarm robotics, is an exciting practical investigation left for future work.
>
> # Generalization
> Our evaluation studies generalization along two axes:
> 1. **Task type**: We train on RAD DFAs and evaluate on Reach and ReachAvoid DFAs, with the same upper bound on the number of states. While Reach and ReachAvoid tasks are technically contained within the RAD class, they differ in transition structure and reward sparsity, inducing a meaningful *distributional shift* &mdash; see Algorithms 1 to 3 and Figure 6 in the Appendix of the paper for more details on these task types.
> 2. **Task size**: We train on RAD DFAs with at most 5 states and evaluate on RAD, Reach, and ReachAvoid DFAs with up to 10 states, producing *out-of-distribution* (OOD) DFAs in terms of their number of states.
>
> We will revise the text to make these axes explicit and better motivate their use.

---

> > ### Author Rebuttal · Reviewer_874m · 2026-04-02
> >
> > My questions have been adequately answered.

---

### Decision · Program_Chairs · 2026-04-30

**Decision:**

Accept (regular)

**Comment:**

This paper proposes Automata-Conditioned Cooperative Multi-Agent Reinforcement Learning (ACC-MARL), a framework for learning multi-task, multi-agent decentralized policies where tasks are represented as DFAs assigned at runtime. The paper identifies three core challenges  (history dependency, credit assignment, and representation bottleneck) and addresses them with a DFA progression for a Markovian reformulation, potential-based reward shaping that preserves optimality, and pretrained frozen RAD embeddings for task representation. Optimality is formally proved, and learned value functions are used to assign tasks at test time by enumerating permutations.

The problem is novel and well-motivated, and it appears to be the first work applying formal specifications in a multi-task MARL setting. The theoretical grounding is solid, the ablation study isolates each contribution, and qualitative analysis demonstrates cooperative behaviors such as door-holding and task short-circuiting.

Some limitations remain in terms of determining how general and scalable the method is. The evaluation is restricted to a custom fully observable discrete environment with teams of up to four agents, and the task assignment procedure does not scale to large teams. The final version should soften the scalability language (add a limitations section as required by ICML), and incorporate the additional discussion of future work and task assignment scalability promised in the rebuttal.